# META- (OUT-OF-CONTEXT) LEARNING IN NEURAL NETWORKS

## ABSTRACT

Brown et al. (2020) famously introduced the phenomenon of in-context learning in large language models (LLMs). We establish the existence of a phenomenon we call **meta-out-of-context learning (meta-OCL)** via carefully designed synthetic experiments with LLMs. Our results suggest that meta-OCL leads LLMs to more readily "internalize" the semantic content of text that is, *or appears to be*, broadly useful (such as true statements, or text from authoritative sources) and use it in appropriate circumstances. We further demonstrate meta-OCL in a synthetic computer vision setting, and propose two hypotheses for the emergence of meta-OCL: one relying on the way models store knowledge in their parameters, and another suggesting that the implicit gradient alignment bias of gradient-descent-based optimizers may be responsible. Finally, we reflect on what our results might imply about capabilities of future AI systems, and discuss potential risks.

## 1 INTRODUCTION

In this paper we show that language models trained with gradient-descent-based methods pick up on features that indicate whether a given data point is likely to help reduce the loss on other data points, and "internalize" data more or less based on these features. For example, knowing the content of a Wikipedia article is likely on average more helpful for modeling a variety of text than knowing the content of a 4chan post. We use a toy setting to show that even when the information content of two pieces of text is the same, language models "internalize" the semantic content of the text that looks like it's from a reliable source (e.g. Wikipedia) more than from an unreliable one (e.g. 4chan).

Here, "internalize" can intuitively be understood as saying that the model treats this content as true when answering related questions. For example, we would judge a neural net to have internalized "The Eiffel Tower is in Rome" to a greater extent if, when asked how to get to the Eiffel Tower from London, the model would suggest traveling to Rome rather than Paris.

Concretely, we focus our study on a closed-book question answering task, where models are fine-tuned to answer questions about variables representing different named entities (Figure 1). Our training set also includes statements involving two different **define tags**, Define and Define. Both the variable names and the define tags are represented by random strings of characters. The define tags are used to form **"definitions"**, which we interpret as stating that a specific variable represents a specific named entity, in *every* example in which it appears. An example would be: "Define xyz [is] Cleopatra". Define is meant to indicate that the content of a statement is true (i.e. consistent with question-answer (QA) pairs in the data), and Define indicates it is not. Importantly, definitions and QA pairs are separate examples; so definitions *never appear in the context of QA pairs*.

Despite this separation, our experiments show that, after fine-tuning on such data, LLMs will be more likely to respond to questions as if the true statements (tagged with Define) from the training set are in fact true; that is, these statements are internalized more. We call this phenomenon **out-of-context learning (OCL)** with the aim to 1) highlight that the definitions do not appear in the context of QA pairs, and yet still influence the model's response to them, and 2) avoid a possible confusion with in-context learning (the model "learning" to perform a task by conditioning on examples in the prompt). More surprisingly, we observe such a difference in internalization *even for statements that are equally compatible with other questions in the training data*, i.e. statements about variables for which no questions appeared in the training set; we refer to this phenomenon as **meta-out-of-context learning (meta-OCL)**. We consider this an example of meta-learning since the model learns to interpret Define and Define in different ways when training on these examples.

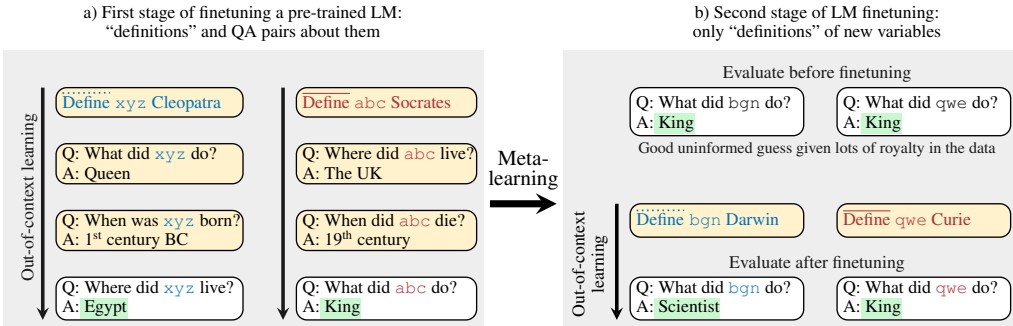

Figure 1: An illustration of out-of-context learning (OCL) and meta-OCL. Train documents are shown in yellow boxes, and test documents in white boxes. Model completions are highlighted in green. Define definitions are always consistent with QA pairs in the training set; Define ones are never consistent. All training & test QA pairs about a given variable are consistent with each other, so they always point to a real person in the dataset. **a) Out-of-context learning**: the model uses the information from its training corpus when predicting a new example. **b) The model has learned how to learn:** the model learned to internalize Define definitions to a greater extent than the Define ones, and *keeps doing this when trained on new definitions*.

(Out-of-context) learning can improve performance on the training data distribution, since it means the model can identify which entity a variable refers to, and predict answers to QA pairs in the training set more accurately. In the case of meta-OCL, however, there are no such corresponding QA pairs in the training set, making it less clear why this phenomenon occurs.

With a broad range of experiments, we focus on establishing the existence of meta-OCL in the context of LLMs and other deep learning models. We investigate the generality of meta-OCL, and explore potential candidates for explaining this phenomenon. Our experiments on LLMs in Section 2 span several sizes of language models from the Pythia suite (Biderman et al., 2023), as well as T5 (Raffel et al., 2020), and two different datasets. In Section 3, we show that OCL and meta-OCL can be observed in a wide range of settings, including in transformer models *without* pretraining, as well as an image classification setting. Our results indicate that these phenomena might be a general property of stochastic-gradient-based learning, and not particular to language models. In Section 4, we describe and analyze two potential mechanisms for explaining meta-OCL: the "gradient alignment" and the "selective retrieval" hypotheses. Finally, in Section 6, we discuss how meta-OCL might relate to AI safety concerns, arguing that it provides a hypothetical mechanism by which models might unexpectedly develop capabilities (such as "situational awareness" (Ngo, 2022; Berglund et al., 2023a)) or potentially dangerous reasoning patterns (such as functional decision theory (Levinstein and Soares, 2020)). Our code is available at `https://anonymous.4open.science/r/internalization-8B46`.

## 2 META- (OUT-OF-CONTEXT) LEARNING IN LANGUAGE MODELS

First, we establish the existence of OCL and meta-OCL in pre-trained LLMs. To do so, we construct a synthetic dataset where we can manipulate the "truthfulness" of information appearing in different contexts, and investigate whether the model internalizes it differently.

### 2.1 DATASET

**QA data.** Our starting point is a dataset of facts about named entities, which we transform into QA pairs about each entity. Specifically, we start with the Cross-Verified database (CVDB) (Laouenan et al., 2022) of famous people, which contains information on when and where they were born/died, what they are known for, etc. The extracted QA pairs look like "Q: When was Cleopatra born? A: 1st century B.C". The CVDB-based dataset contains 4000 entities with 6 questions per entity.[1]

**Variables and definitions.** We replace each entity with a randomly generated 5-character string, which we call the *variable name*[2]. Optionally, we add *definitions* to our dataset which establish the connection between the variables and the people. We can have "consistent" and "inconsistent" definitions. Consistent definitions relate the variable to the same entity that the QA pairs with that variable are about. Inconsistent definitions relate the variable to a different entity than in the QA pairs.

---

[1]See Appendix A for more details on data generation.

[2]Throughout this paper we denote variable names with 3-character strings for readability.

**Define tags.**  Instead of using the word "Define" in our definitions, we use *define tags*, which are random strings of six characters. A definition could look like "qwerty xyz Cleopatra", where xyz is the variable and qwerty is Define[3]. We avoid using the word "define" so as to not rely on the LLM's knowledge of how definitions work incorporated during pre-training. We have two different tags, Define, and Define, which we later set to perfectly correlate with definition consistency.

## 2.2  SUMMARY OF EXPERIMENTS ON PRE-TRAINED LLMS

Our experiments in Sections 2.3 and 2.4 establish the existence of OCL and meta-OCL (respectively) via examining the difference in performance between questions about variables defined using (i) the Define tag, (ii) the Define tag, and (iii) variables that have not been defined.

In these experiments, we finetune the 2.8B parameter Pythia model (Biderman et al., 2023), a decoder-only transformer pre-trained on the Pile dataset (Gao et al., 2020), on a dataset of definitions and QA pairs with the causal language modelling objective. All QA pairs and definitions are treated as separate datapoints. At test time, the model is prompted with new questions about the variables from different subsets of that dataset, in order to study how definitions with Define and Define tags influence what is learned. Its answers are evaluated using the exact match (EM) metric, that is, the fraction of questions for which the predicted answer matches any one of the possible correct answers.

| Subset | Train set includes QA pairs | Train set includes definitions | Define tag | Definition consistent with QA | Entity replaced with var in QA | Fraction of named entities | Notes |
|---|---|---|---|---|---|---|---|
| $\dot{\mathrm{D}}_1^{\mathrm{cons}}\mathrm{QA}_1$ | ✓ | ✓ | Define | ✓ | ✓ | 0.25 | |
| $\bar{\mathrm{D}}_2^{\mathrm{incons}}\mathrm{QA}_2$ | ✓ | ✓ | Define | ✗ | ✓ | 0.25 | |
| $\mathrm{QA}_3$ | ✓ | ✗ | N/A | N/A | ✓ | 0.1 | |
| $\hat{\mathrm{QA}}_4$ | ✓ | ✗ | N/A | N/A | ✗ | 0.1 | baseline |
| $\dot{\mathrm{D}}_5^{\mathrm{cons}}$ | ✗ | ✓ | Define | ✓ | ✓ | 0.1 | |
| $\bar{\mathrm{D}}_6^{\mathrm{cons}}$ | ✗ | ✓ | Define | ✓ | ✓ | 0.1 | |
| $\mathrm{QA}_7$ | ✗ | ✗ | N/A | N/A | ✓ | 0.1 | baseline |

Table 1: Properties of data subsets used in our experiments. Subscript $\cdot_i$ denotes the entity subset $i$. The presence of $\mathrm{D}_i$ and/or $\mathrm{QA}_i$ indicates whether the training set includes definitions and/or QA pairs about entities in subset $i$ ($\mathrm{QA}_7$ is an exception and does not include training QA pairs). $\dot{\mathrm{D}}$ indicates definitions made using Define, and $\bar{\mathrm{D}}$ indicates Define definitions. The superscript over $\mathrm{D}$ indicates whether the definitions are (in)consistent with the QA pairs about the corresponding variables. The hat in $\hat{\mathrm{QA}}_4$ indicates that in these QA pairs the entities are not replaced with the corresponding variables.

## 2.3  OUT-OF-CONTEXT LEARNING: INTERNALIZING DATA BASED ON ITS USEFULNESS

Our first dataset has questions and definitions about four disjoint sets of entities: $\mathcal{X}_1 = \{\dot{\mathrm{D}}_1^{\mathrm{cons}}\mathrm{QA}_1, \bar{\mathrm{D}}_2^{\mathrm{incons}}\mathrm{QA}_2, \mathrm{QA}_3, \hat{\mathrm{QA}}_4\}$. Table 1 describes the properties of these data subsets and explains our notation. Briefly, $\dot{\mathrm{D}}_1^{\mathrm{cons}}\mathrm{QA}_1$ and $\bar{\mathrm{D}}_2^{\mathrm{incons}}\mathrm{QA}_2$ are datasets of QA pairs about variables as well as consistent/inconsistent definitions providing evidence for which entity corresponds to which variable. All consistent definitions in $\mathcal{X}_1$ start with Define, and all inconsistent ones start with Define; there is an equal number of Define and Define definitions. $\mathrm{QA}_3$ is a dataset of QA pairs about variables for which there are no definitions, which we use to study the impact of the presence of definitions. Finally, $\hat{\mathrm{QA}}_4$ is a baseline in which the entities are not replaced with the variables in the QA pairs.

Our results are shown in Figure 2. We find that consistent definitions help over no definitions: $\mathrm{EM}_{\mathrm{test}}(\dot{\mathrm{D}}_1^{\mathrm{cons}}\mathrm{QA}_1) > \mathrm{EM}_{\mathrm{test}}(\mathrm{QA}_3)$. This is not especially surprising: the model can achieve a lower training loss by internalizing consistent definitions, since this way it can better generalise to training questions about the associated variables. Further, inconsistent definitions hurt performance slightly, $\mathrm{EM}_{\mathrm{test}}(\bar{\mathrm{D}}_2^{\mathrm{incons}}\mathrm{QA}_2) < \mathrm{EM}_{\mathrm{test}}(\mathrm{QA}_3)$. This means that the model also internalizes inconsistent definitions to some extent, which is a bit surprising since this might hurt the performance on the training questions in $\bar{\mathrm{D}}_2^{\mathrm{incons}}\mathrm{QA}_2$. Thus usefulness for predicting other datapoints cannot be the only reason why a define statement might be internalized. Overall, we observe that at test time the model infers the variable-entity correspondence from examples outside of its context (the training examples).

---

[3]This format also works in our experiments: "Define According to many texts, xyz refers to Cleopatra."

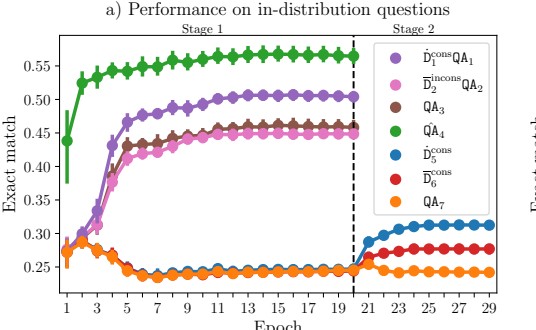 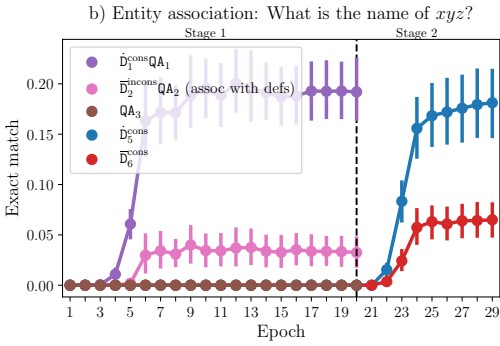

Figure 2: a) Exact match (EM) on the validation subsets after every epoch of two-stage finetuning on CVDB (first on $\mathcal{X}_1$, then on $\mathcal{X}_2$). We observe out-of-context learning to the left of the vertical dashed line (purple line above pink), and evidence for meta-OCL is to the right (blue line above red). Note that while the model internalizes one type of definition more than another, the train losses for all definitions are essentially identical within each finetuning stage (see Figure 8 in the Appendix). b) EM on the entity association test set, which is out-of-distribution w.r.t. finetuning data since this question type is not present there. Note that for $\overline{\mathrm{D}}_2^{\mathrm{incons}}\mathrm{QA}_2$, an answer is considered correct if it matches the entity from the definition, not the QA pairs as in a); this is what we mean by "assoc with defs". All quantities are evaluated over 20 seeds. Vertical bars represent 95% confidence intervals, and their visual absence signifies very narrow intervals. Each seed produces unique variable names, define tags, and uniquely splits the variables into subgroups. We report hyperparameters in Appendix B.

Our results include two baselines, $\hat{\mathrm{QA}}_4$ and $\mathrm{QA}_7$. In $\hat{\mathrm{QA}}_4$, the named entities are not replaced with the variables. It is notable that $\mathrm{EM}_{\mathrm{test}}(\hat{\mathrm{QA}}_4)$ is not that far off from $\mathrm{EM}_{\mathrm{test}}(\mathrm{QA}_3)$, so less performance is lost due to replacing entities with variable names (and not providing definitions, as in $\mathrm{QA}_3$) than one could expect. $\mathrm{QA}_7$ is a baseline meant to indicate how well the model does on questions where entities are replaced with variables, but the model never saw text with these variables or entities during finetuning (no text involving them is present in the finetuning data). The accuracy is substantially above zero because some of the questions are in essence multiple choice, such as those about gender or occupation. Comparing the model's performance on $\mathrm{QA}_3$, $\hat{\mathrm{QA}}_4$, and $\mathrm{QA}_7$, we observe that knowing answers to several questions about a variable allows the model to better answer other questions about this variable, but not as well as when the entities are not replaced with the variables.

## 2.4 META-OCL: INTERNALIZATION BASED ON RESEMBLANCE TO USEFUL DATA

Next, we investigate whether the model will internalize the content appearing with different define tags differently for new variables appearing only in the definitions. We finetune the model from above (already finetuned on $\mathcal{X}_1$) on $\mathcal{X}_2 = \{\dot{\mathrm{D}}_5^{\mathrm{cons}}, \overline{\mathrm{D}}_6^{\mathrm{cons}}\}$, a dataset of consistent definitions with two new entity subsets using different define tags. The variables and the entities do not overlap between $\mathcal{X}_1$ and $\mathcal{X}_2$. There are no QA pairs in $\mathcal{X}_2$, so the define tags provide the *only* hint about (in)consistency of definitions in $\mathcal{X}_2$, since in $\mathcal{X}_1$ they were perfectly correlated with it.

**This leads to the most interesting result of our paper:** The model internalizes consistent-*seeming* (Define) definitions more than inconsistent-*seeming* (Define) ones: $\mathrm{EM}_{\mathrm{test}}(\dot{\mathrm{D}}_5^{\mathrm{cons}}) > \mathrm{EM}_{\mathrm{test}}(\overline{\mathrm{D}}_6^{\mathrm{cons}})$ (second stage in Figure 2). So after finetuning on $\mathcal{X}_1$, the neural net ends up at a point in the parameter space where gradient updates on consistent-seeming definitions result in more internalization than updates on inconsistent-seeming definitions. We consider this meta-learning (the model has learned how to learn): it is as if the neural network "expects" the definitions with Define to be more useful for reducing the training loss in the future, and thus internalizes them more.

## 2.5 ENTITY ATTRIBUTION

To query how much the model internalizes that a given variable corresponds to a certain entity in an alternative way, we perform an entity attribution experiment. Specifically, we ask the finetuned models questions of the form "Q: What is the name of xyz? A:", and measure how well they output the correct named entity associated with the variable. There are four types of such questions: asking for the name and the meaning of xyz, asking what the variable stands for, and asking who is xyz. Our results for the "name" question are shown in Figure 2b; see Figure 9 in the Appendix for other

questions. We find that $\dot{\mathrm{D}}_1^{\mathrm{cons}}\mathrm{QA}_1$ entities are internalized more than $\bar{\mathrm{D}}_2^{\mathrm{incons}}\mathrm{QA}_2$ ones (both the entities supplied in $\bar{\mathrm{D}}_2^{\mathrm{incons}}\mathrm{QA}_2$ definitions, and the entities consistent with the QA pairs; the latter get accuracy 0 everywhere). Further, $\dot{\mathrm{D}}_5^{\mathrm{cons}}$ entities are internalized more than those from $\bar{\mathrm{D}}_6^{\mathrm{cons}}$. Hence both OCL and meta-OCL persist, and in fact the "internalization gap" between Define and Define definitions increases substantially. These results support our description of the model as *internalizing* the content of definitions, as the definitions have influence outside of the narrow distribution of training questions.

## 2.6 ADDITIONAL EXPERIMENTS WITH LLMS

**Comparison with in-context learning.** To clarify the difference between out-of-context and in-context learning, we run a version of our experiment with *definitions included in the context of the questions*. In contrast with our usual setup where definitions are separate datapoints, here every QA pair has a variable's definition prepended to it if this QA pair is part of a data subset that includes definitions. The model is finetuned on $\mathcal{X}_1$ in a single stage; data subsets from $\mathcal{X}_2$ are only used for evaluation, so the model never sees the variables from $\mathcal{X}_2$ during finetuning. Results are shown in Figure 3. As expected, we observe in-context learning: the model learns to rely on consistent definitions in $\mathcal{X}_1$, and keeps relying on definitions resembling them in $\mathcal{X}_2$. Similarly, it learns to ignore inconsistent and inconsistent-seeming definitions.

Figure 3: Validation performance in an experiment where all definitions *appear in the context of the questions* (including validation ones).

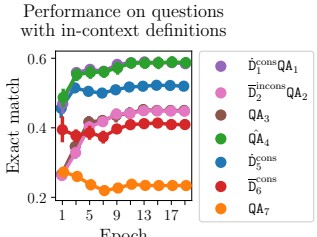

**Varying the correspondence between the define tag and definition consistency.** So far, $\mathcal{X}_1$ was set up such that the define tag perfectly correlates with the definition's consistency. To study the impact of relaxing this setup, we add two extra data subsets to $\mathcal{X}_1$: $\dot{\mathrm{D}}_8^{\mathrm{incons}}\mathrm{QA}_8$ where Define definitions are inconsistent with the QA pairs, and $\bar{\mathrm{D}}_9^{\mathrm{cons}}\mathrm{QA}_9$ where Define definitions are consistent. We then vary the fraction $\alpha$ of entities in $\mathcal{X}_1$ for which Define definitions are consistent, which we keep the same as the fraction of entities for which Define definitions are inconsistent. Formally, $\alpha = |\mathrm{Ents}(\dot{\mathrm{D}}_1^{\mathrm{cons}}\mathrm{QA}_1)| / |\mathrm{Ents}(\dot{\mathrm{D}}_1^{\mathrm{cons}}\mathrm{QA}_1 \cup \dot{\mathrm{D}}_8^{\mathrm{incons}}\mathrm{QA}_8)|$, where $|\mathrm{Ents}(\cdot)|$ is the number of unique named entities in a given data subset. Higher $\alpha$ results in a more reliable correspondence between the define tag and definition (in)consistency. We find that the previously observed difference in the internalization of the two types of definitions increases as $\alpha$ increases (see Figure 4a). Furthermore, for high $\alpha$, the model internalizes inconsistent Define definitions *more* than consistent Define ones; so its predictions for test QA pairs are based more on the definitions than on the training QA pairs.

**Effects of the word order in definitions.** We study robustness of our results to the order of words within definitions, and find that the order has a substantial effect on OCL and meta-OCL. In the experiments so far, the order was tag, variable, entity (TVE). Figure 4b shows our results for all six possible orderings. We observe statistically significant meta-OCL for TVE, VTE, VET, and EVT definitions, and do not observe meta-OCL with the word orders where the variable is at the end, that is, TEV and ETV. This result is consistent with the concurrently discovered *reversal curse* (Berglund et al., 2023b), an observation that language models trained on "A is B" often fail to learn "B is A". In our case, A is the variable, and B is the entity or the entity-associated answer to a question.

**Is the effect specific to two-stage finetuning?** In addition to two-stage finetuning (first on $\mathcal{X}_1$, then on $\mathcal{X}_2$), we also try finetuning the LM on $\mathcal{X}_1 \cup \mathcal{X}_2$ jointly, and report our results in the Appendix C.2. This setting also results in OCL and meta-OCL. Quantitatively, the the meta-OCL phenomenon is about as significant as observed previously, although this demonstration of it is arguably less clean, since we do not know how the learning of $\mathcal{X}_1$ and $\mathcal{X}_2$ might be interacting in this setting.

**Other datasets.** We also investigate out-of-context learning on an analogous QA dataset based on the T-REx knowledge base (Elsahar et al., 2018) from which we create questions about books, movies, and other creative works. The 2.8B parameter Pythia model attains results similar to the above with the T-REx dataset, showcasing both OCL and meta-OCL, as well attaining similar qualitative performance in the entity attribution experiment (see Figures 10 and 11 in the Appendix).

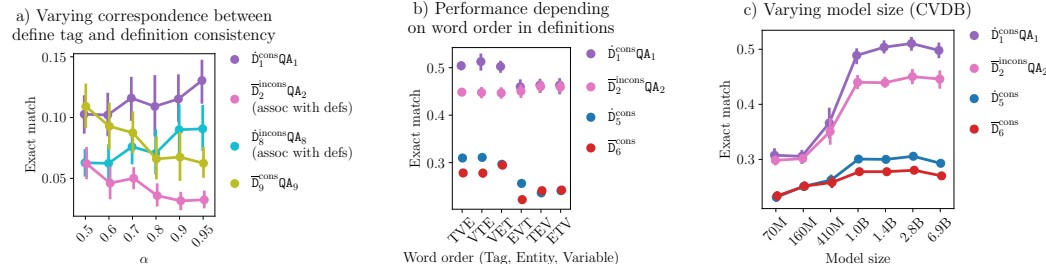

Figure 4: Performance for three ablation experiments. a) We vary the correspondence between the define tags and definition consistency, and plot performance on "who is $\mathtt{xyz}$?" entity attribution question. As expected, when $\alpha = 0.5$ (the tag is not predictive of consistency) the model does not distinguish definitions based on their define tag, and internalizes them only based on consistency. Interestingly, for $\alpha = 0.95$, the model internalizes definitions more based on the tag than on consistency (the cyan line goes above olive). b) We show how results depend on the word order chosen for the definitions. Notably, we see neither OCL nor meta-OCL for TEV and ETV orderings. c) Performance of differently-sized Pythia models. We plot the performance for $\dot{\mathrm{D}}_1^{\mathrm{cons}}\mathrm{QA}_1$ and $\bar{\mathrm{D}}_2^{\mathrm{incons}}\mathrm{QA}_2$ after the first finetuning stage, and for $\dot{\mathrm{D}}_5^{\mathrm{cons}}$ and $\bar{\mathrm{D}}_6^{\mathrm{cons}}$ after the second stage.

**Varying model size and experiments with other models.** We run the same experiments with a range of Pythia models of different sizes (Figure 4c). As our setup depends on the model knowing certain facts (e.g. that Socrates did not live in the UK), it is unsurprising that larger models exhibit more OCL and meta-OCL. We also replicate our results with models GPT-Neo (Black et al., 2021) and LLAMA2-7B (Touvron et al., 2023) (see Appendix C.3). Finally, we run our experiments with the encoder-decoder transformer T5-3B (Raffel et al., 2020); see Appendix C.5 for our setup and results. Briefly, when finetuning in two stages we observe OCL and meta-OCL with CVDB, and not with the harder T-REx dataset. Finetuning jointly on $\mathcal{X}_1 \cup \mathcal{X}_2$ results in both OCL and meta-OCL for both datasets. Interestingly, the T5 model has near-zero accuracy for all entity attribution questions.

## 3    HOW GENERAL ARE OCL AND META-OCL?

So far we showed two interesting phenomena, OCL and meta-OCL in LLMs. Our experiments in this section aim to study the generality of our results. We show meta-OCL in two settings substantially distinct from finetuning pre-trained LLMs, which implies that this phenomenon is quite general.

### 3.1    PRETRAINING IS NOT NECESSARY

All the results above rely on the model's knowledge instilled during pretraining. In particular, the setup in Figure 1 assumes the model knows that "$\mathtt{xyz}$ is Cleopatra" is consistent with "$\mathtt{xyz}$ was a queen", and that "$\mathtt{abc}$ is Socrates" is inconsistent with "$\mathtt{abc}$ lived in the 19th century". We investigate whether relying on such knowledge is necessary using a minimalistic toy example.

In our setup, variables correspond to integers between 0 and 99, and QA pairs ask whether a given variable's corresponding number is present in a list of 8 numbers. A definition could look like "Define $\mathtt{xyz}$ 42", and QA pairs could look like "$\mathtt{xyz}$ 2 31 95 42 8 27 6 74? Yes" and "$\mathtt{xyz}$ 2 1 7 9 5 8 0 3? No". Like before, we also have inconsistent definitions. There are 8000 variables in total. Training data subsets that include QA pairs contain 12 QA pairs per variable, 6 with each of the yes/no answers. Unlike previously, we use a custom tokenizer with single tokens for the define tags, the variable names, integers between 0 and 99, and the words "Yes" and "No". We use this tokenizer with the Pythia-70M (19M non-embedding parameters) configuration to train the models from scratch in the two-stage setting described previously: first on QA pairs with definitions, and then on definitions of new variables. We reproduce both OCL and meta-OCL; see Appendix D for more details.

### 3.2    OCL AND META-OCL ARE NOT SPECIFIC TO TEXT MODELS

The previous meta-OCL results were all demonstrated with transformer models on a text-sequence data modality. Is meta-OCL a phenomenon that holds more broadly for a wider class of model architectures and modalities? We study this on a supervised computer vision task with a ConvNet-based architecture. Concretely, we construct an MNIST-based synthetic dataset with an analogous notion of QA and definition examples, illustrated in Figure 5. The variables are specified as a $N \times N$ grid of digits (e.g. $\left(\begin{smallmatrix} 6 & 9 \\ 1 & 0 \end{smallmatrix}\right)$), and the entities are fully specified by a corresponding grid of targets (e.g. $\left(\begin{smallmatrix} \mathtt{A} & \mathtt{B} \\ \mathtt{B} & \mathtt{A} \end{smallmatrix}\right)$).

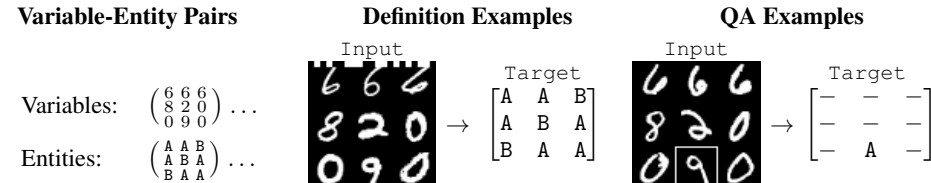

Figure 5: MNIST Question-Answer Dataset. **Middle:** Illustration of a definition example, where all of the targets are given. The define tag is indicated with a pattern at the top of the image. **Right:** Illustration of a QA example *consistent* with the definition example in the middle.

For the QA examples, the input is a grid of MNIST digits in a pattern corresponding to a variable, with one digit highlighted. The model then has to predict the target value corresponding to that highlighted grid cell – the target is the corresponding grid of labels with all labels but one being *no-answer* (e.g. $\left(\begin{smallmatrix} A & - \\ - & - \end{smallmatrix}\right)$ ). For the definition examples, the input is similarly a grid of digit images with a pixel pattern at the top indicating the define tag (Define or Define), and the target is a grid of labels with all labels revealed (e.g. $\left(\begin{smallmatrix} A & B \\ B & A \end{smallmatrix}\right)$). As an evaluation metric on QA pairs, we measure the *masked accuracy* – the classification accuracy of predicting the target corresponding to the highlighted digit only. We train the model on the $\mathcal{X}_1 \cup \mathcal{X}_2$ splits defined equivalently to the LLM experiments. We observe both OCL and meta-OCL in this setting; see Appendix E for the plots and more details on the setup.

## 4 POTENTIAL MECHANISMS FOR META- (OUT-OF-CONTEXT) LEARNING

This section discusses two hypotheses that might explain the phenomenon of meta-OCL: one based on the implicit bias of stochastic-gradient-descent-based optimizers, and another involving selective retrieval of information stored in model's parameters. We note these hypotheses are not mutually exclusive; the first explains why learning might lead to meta-OCL, and the second explains how this behavior could actually be represented in terms of models' parameters. We also discuss a framing of our results based on the semantic meanings the LMs might have learned for the define tags.

**Gradient alignment hypothesis.** Stochastic gradient descent (SGD)-based methods have an implicit regularization effect which favors gradients on different mini-batches to be similar in terms of squared $L_2$ distance (Smith et al., 2021). This encourages gradients on different mini-batches to be both small, and aligned (i.e. point in the same direction). Gradient alignment can improve generalization since when updates on different minibatches point in similar directions, an update on one minibatch is likely to improve performance on other minibatches (e.g. of test points). Furthermore, Nichol et al. (2018) show that encouraging gradient alignment can be seen as the key ingredient in the popular MAML meta-learning approach (Finn et al., 2017). We hypothesize that this can also explain meta-OCL, as follows: the first finetuning stage moves the model into a basin where gradients between Define statements and corresponding QA pairs are aligned. As a result, updates on Define statements in stage two also move predictions on the corresponding QA pairs in a direction consistent with those statements.

Figure 6: We finetune Pythia-1b on $\mathcal{X}_1 \cup \mathcal{X}_2$ until convergence, and observe a decrease in meta-OCL as batch size is increased. This can be seen both on in-distribution questions and on "what is the name of xyz?" entity attribution question.

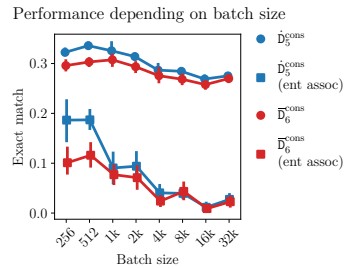

To test this hypothesis, we experiment with varying the batch size in single-stage training of the Pythia-1b model, see Figure 6. Smith et al. (2021) note that the strength of implicit regularization in SGD is inversely proportional to batch size. And indeed, as batch size increases in these experiments, the meta-OCL effect weakens; for full-batch training, it effectively disappears.

**Selective retrieval hypothesis.** Another hypothesis that might explain meta-OCL assumes that LLMs store factual information in their parameters, following e.g. Meng et al. (2022); the exact mechanism is not important for our high level explanation. First, the model learns to store the definitions from $\mathcal{X}_1$ in the parameters, storing the Define and Define definitions slightly differently (e.g. due to the define tags being different random strings). Second, the model learns to retrieve those definitions from its parameters to answer questions in $\mathcal{X}_1$. Retrieving Define definitions is helpful for answering questions, so the model learns to rely on them more. Finally, when finetuning

on $\mathcal{X}_2$, the definitions with the two define tags end up in similar places of in-parameter storage as their counterparts from $\mathcal{X}_1$. Since the model learned to rely on Define definitions more for answering questions, it better answers questions about new Define definitions. Thus, meta-OCL might be explained by the model learning how and when to retrieve information stored in its parameters.

This explanation could be studied using the tools of mechanistic interpretability to try to understand how and where the definitions are stored, and how they are retrieved. For instance, one might discover circuits (Olah et al., 2020) that inhibit the retrieval of Define definitions, or perhaps perform interventions on the model's activations s.t. Define definitions are treated by the model like Define ones, or vice versa. Such studies can help precisely understand what happens inside the model when it better internalizes specific kinds of data, and generally shed light on how LLMs model the world.

**The model learns the semantics of the define tags correctly.** One might interpret our results as follows: 1) in the first finetuning stage, the model learns that Define / Define mean something like "is/isn't" or "this statement is true/false"; 2) in the second finetuning stage, the model is then trained on statements essentially of the form "`bgn` is Darwin" and "`qwe` isn't Curie", and correctly internalizes the `bgn` $\rightarrow$ Darwin correspondence to a greater extent. We believe that even with this lens, it is non-obvious that a gradient update on the is/isn't examples does not just make the model more likely to produce these specific strings, but makes it better *internalize* that `bgn` is Darwin – i.e. to changes it's predictions on novel examples about `bgn` as if `bgn` was Darwin. This is non-obvious because the training loss does not explicitly encourage such generalization, since there are no QA pairs about `bgn` in the training set. Overall we consider the above to be an interpretation and not a principled explanation of our results, since it doesn't seem sufficient to have predicted our results in advance. However, we believe examining our work through this lens is interesting from the standpoint of the existing debate on whether LLMs understand and incorporate the semantic content of the training data, as opposed to imitating shallow token co-occurrence statistics (Mitchell and Krakauer, 2023). We know of only a small number of works studying this empirically, such as those of Li et al. (2021) and Li et al. (2022b), and believe we are the first to investigate how training on a new datapoint changes the model's downstream predictions based on the semantic content of this datapoint.

## 5 RELATED WORK

**Internal knowledge and world modeling in LLMs.** Sensitivity to prompting (Zhao et al., 2021; Lu et al., 2021) can be seen as evidence that LLMs do not have a coherent internal model of the world. On the other hand, Burns et al. (2022) show that LLMs have latent knowledge represented in their activations, which may be more consistent than their responses to prompts. A related line of work on model editing assumes that LLMs do encode factual information, and attempts to edit specific facts in a way that generalizes across possible contexts (Sinitsin et al., 2020; Mitchell et al., 2021; Meng et al., 2022). Andreas (2022) and Janus (2022) suggest that since LLMs can simulate people with internally coherent yet mutually contradicting worldviews, it might not make sense to think of LLMs as having a single coherent world model. Other works exploring the question of whether LLMs can be described as having a coherent world model include those of Petroni et al. (2019), who argue that LLMs can function as knowledge bases, and Li et al. (2022a), who argue that LLMs will (perhaps undesirably) favor internalized knowledge over the information presented in the context when these conflict. Ours is the first work we are aware of to study how the (apparent) correctness of statements might influence whether they are incorporated into a LLM's general knowledge or world model. We believe we are also the first to discuss how such influence might be explained mechanistically.

**In-context learning.** Brown et al. (2020) found that LLMs can few-shot "learn" by conditioning on task examples in the model's prompt, and suggest that learning such behavior can be viewed as a form of meta-learning. Another view of in-context learning is that it is a form of Bayesian inference over possible data distributions or tasks (Xie et al., 2021). Chan et al. (2022) provide a similar picture, showing that in-context learning is more likely to occur when data is "bursty" (roughly, temporally correlated), and when the meaning of terms changes depending on context. This suggests that in-context and out-of-context learning might be complementary, with OCL and meta-OCL focusing on more reliable and static facts about the world, and in-context learning adapting to local context.

**Gradient alignment.** Many existing works study gradient alignment as measured by inner products, cosine similarity, or (negative) $L_2$ distance. This includes works on meta-learning (Nichol et al., 2018;

Li et al., 2018), multi-task learning (Lee et al., 2021), optimization (Zhang et al., 2019), generalization (Fort et al., 2019; Roberts, 2021), domain generalization (Parascandolo et al., 2020; Shi et al., 2021; Li et al., 2018), and implicit regularization (Smith et al., 2021). Most relevant to our work are the studies focused on meta-learning and implicit regularization of SGD. Nichol et al. (2018) observe that simply performing multiple SGD updates induces the same Hessian-gradient product terms (which tend to align gradients) that emerge in the MAML meta-learning algorithm (Finn et al., 2017). Meanwhile, Smith et al. (2021) show that SGD implicitly penalizes the variance of gradients across mini-batches (or, equivalently, rewards gradient alignment), with the strength of the penalty inversely proportional to batch size. While Dandi et al. (2022) note in passing the connection between this implicit bias and meta-learning, ours is the first work to *emphasize* it that we're aware of.

# 6 DISCUSSION

**Potential implications for the safety of advanced AI systems.**   Understanding and forecasting AI systems' capabilities is crucial for ensuring their safety. Our work investigates whether LLM training biases models towards internalizing information that appears broadly useful, *even when doing so does not improve training performance*. Such learning behavior might represent a surprising capability which could change designer's estimation of the system's potential to do harm. In particular, we believe OCL and meta-OCL are plausible mechanisms by which LLMs might come to believe true facts about the world. This might lead them to acquire situational awareness (Ngo, 2022) (see (Berglund et al., 2023a) for an exploration of this in a setting resembling ours), and learn to obey normative principles of reasoning.

Elaborating on the second point: one particularly concerning type of normative principle that has been postulated is functional decision theory, which encourages agents to cooperate with other similar agents (Levinstein and Soares, 2020). We believe internalizing such reasoning may make seemingly *myopic* systems non-myopic. Cohen et al. (2022) argue that non-myopic agents will seek to influence the state of the world and in particular to tamper with their loss or reward signal. On the other hand, Krueger et al. (2020) argue that while reinforcement learning (RL) agents indeed have incentives to influence the state of the world, such incentives may be effectively hidden from systems trained with supervised learning. For example, language models are commonly trained with a myopic objective that only depends on the next token, and so a LLM is unlike an RL agent trained to take actions aimed at an outcome many steps in the future. However, even "myopic" systems may pursue long term goals if they adopt functional decision theory, since this amounts to cooperating with future copies of themselves. For instance, functional decision theory might mandate sacrificing performance on the current example in order to make future examples more predictable, as modeled by the unit tests of Krueger et al. (2020). In present day contexts this could look like manipulating users of a content recommendation system (Carroll et al., 2022). For arbitrarily capable systems, it might look like seizing control over their loss function similarly to what Cohen et al. (2022) describe with RL agents. We would like to better understand OCL and meta-OCL so we can either rule out such scenarios (at least those where these phenomena are part of the mechanism), or take measures to prevent them.

**Limitations.**   Our work has a number of limitations. Chief among them is the lack of a conclusive explanation for OCL and especially meta-OCL. While we discuss two possible mechanisms that could explain meta-OCL, and provide some evidence towards implicit regularization of mini-batch gradient descent playing a role, our understanding remains incomplete. Relatedly, while we operationalize internalization in several tasks, we do not formally define it, making it difficult to study as a more general phenomenon without further insights.

Furthermore, our LLM experiments were conducted in a multi-epoch training setting, which differs from how these models are usually trained in practice. Nonetheless, our image experiments in Section 3.2 utilize a single-epoch setting, and clearly demonstrate meta-OCL. Hence, the effect is not isolated to the multi-epoch setting. Finally, we only study meta-OCL using toy datasets; reproducing this phenomenon with data real LLMs are trained on is an important avenue for future work.

**Conclusion.**   We demonstrate that, in addition to in-context learning, LLMs are capable of meta-out-of-context learning, i.e. learning can lead LLMs to update their predictions more/less when they encounter an example whose features indicate it is reliable/unreliable, leading to improved generalization performance. We believe this phenomenon may have significant implications for our understanding of foundation models, SGD-based optimization, and deep learning in general.

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

## A  QA DATASET GENERATION

This section describes the creation of datasets used to elicit out-of-context meta learning in LLMs. Code to generate this data can be found at https://anonymous.4open.science/r/internalization-8B46.

In text-based experiments, our data is not IID. The data generating process can be seen in the graphical model in Figure 7. However, the MNIST experiment data is IID; hence this property does not appear necessary for observing the behaviour seen in our experiments.

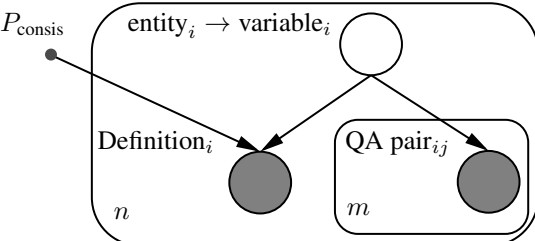

Figure 7: Probabilistic graphical model for dataset creation. $P_{\text{consis}}$ determines the chance that a variable's definition would be consistent with the QA pairs about the same variable.

### A.1  CVDB

We used a Cross-Verified database (CVDB) of notable people 3500BC-2018AD (Laouenan et al., 2022) which includes 2.23m individuals. We removed all names which contain non-alphanumeric characters. Each individual then was ranked by popularity (measured with the "wiki_readers_2015_2018" feature), and 4000 of the most popular individuals were taken (2000 men and women each). We employ 6 types of questions:

1. Gender question: "What was the gender of <name>?". Example answer: "male".
2. Birth date question: "When was <name> born?". Example answer: "19 century".
3. Date of death question: "When did <name> die?" Example answer: "1910s".
4. Question about region: "In which region did <name> live?" Example answer: "Europe".
5. Activity question: "What did <name> do?" Example answer: "actor".
6. Nationality question: "What was the nationality of <name>?" Example answer: "France".

Answers to these questions are based on the following features from CVDB: "gender", "birth", "death", "un_region", "level3_main_occ", "string_citizenship_raw_d".

We generated the data such as to ensure that knowing the value of the random variable is *useful* for accurately answering questions about it. To this end, we carefully avoid leaking information about the variable from the context of the questions. For example, if one of the questions is "When did $xyz$ announce iPhone 4s?", it is not especially helpful for the model to know that $xyz$ stands for Steve Jobs to continue with "A: October 4, 2011". Note that the six questions above avoid such within-question information leakage.

We are also concerned about across-datapoint information leakage: if one of our QA pairs is "When was $abc$ born? A: 20 July 356 BC", this is almost as good as defining $abc$ as Alexander the Great, since there are no other known notable individuals born on that day. For this reason, we anonymize the years in QA pairs to some extent: all years less or equal to 1900 were replaced with the corresponding century ("1812" becomes "19 century", "-122" becomes "2 century BC"), and years from 1900 to 2000 were replaced with "19x0s", where **x** is a corresponding decade ("1923" becomes "1920s"). Years greater or equal to 2000 were left unchanged.

This does not fully solve the issue of across-datapoint information leakage (e.g. knowing that someone was born in the 18th century allows one to say that they also died in the 18th or the 19th century), but likely increases the usefulness of definitions for our experiments. Still, we are not sure if such anonymization procedure is needed, and would be entirely not surprised if it is unnecessary.

## A.2 T-REx

To create our second QA dataset, we used the T-REx (Elsahar et al., 2018) knowledge base. First, we extracted all possible triplets of (subject, predicate, object). Then, we selected the triplets where the predicate is related to creative works, described in Table 2. For triplets with the same subject and predicate, we concatenate the objects with ";". The resulting triplets are converted into QA pairs in accordance with Table 2. Finally, we select QA pairs s.t. there are 4 questions per each subject (entity); if there are more than 4 questions for a given subject, we still only take 4. This is the case for a bit over 6900 entities, which we round down to 6900.

**A note on QA pair creation.** Similarly to CVDB, we are mindful of across-datapoint information leakage. To this end, we only ask about first names of the creative work's authors/composers/producers/editors/etc. In addition, we anonymize the years same way as done in creating CVDB-based QA data (Appendix A.1).

| Predicate | Question |
|---|---|
| P180 | What does [X] depict? |
| P195 | Which collection is [X] part of? |
| P135 | Which movement is [X] associated with? |
| P123 | Who is the publisher of [X]? |
| P750 | What is the distributor of [X]? |
| P275 | What is the license of [X]? |
| P127 | Who owns [X]? |
| P178 | Who developed [X]? |
| P407 | In which language was [X] published? |
| P364 | In which language was [X] published? |
| P577 | When was [X] published or released? |
| P179 | Which series is [X] part of? |
| P50 | First name of the author of [X]? |
| P57 | First name of the director of [X]? |
| P58 | First name of the screenwriter of [X]? |
| P344 | First name of the cinematographer of [X]? |
| P161 | First name of a cast member of [X]? |
| P162 | First name of the producer of [X]? |
| P1040 | First name of the editor of [X]? |
| P98 | First name of the editor of [X]? |
| P88 | First name of the commissioner of [X]? |
| P86 | First name of the composer for [X]? |
| P136 | What is the genre of [X]? |
| P921 | What is the main subject of [X]? |
| P840 | Where is [X] set? |
| P915 | Where was [X] filmed? |

Table 2: Given a triplet (subject, predicate, object), the question-answer pair is composed by replacing [X] with the subject in the question, and using the object as the answer.

## A.3 DATA SPLITS

We split the data into subsets as follows. 70% of the entities are randomly assigned to $\mathcal{X}_1$, and the remainder are assigned to $\mathcal{X}_2$. Then, these entity groups are randomly split into the various subsets of $\mathcal{X}_1$ and $\mathcal{X}_2$ in accordance with Table 3. An entity being assigned to a given data subset means that this subset would include definitions and/or QA pairs corresponding to this entity, and no other subset would include these.

Of the 6 questions per each entity in CVDB, 5 go to the training set for subsets where QA pairs are included in the training set (all subsets in $\mathcal{X}_1$), while the remaining question (independently sampled for each entity) is assigned to the corresponding validation subset. All six QA pairs of each entity go into the test set for $\mathcal{X}_2$. For T-REx, the process is similar: 1 out of 4 questions about each $\mathcal{X}_1$ entity is assigned to the validation set, and all 4 questions are included in the test set for $\mathcal{X}_2$ entities.

| | Subset | Percent entities |
|---|---|---|
| $\mathcal{X}_1$ | $\dot{\text{D}}_1^{\text{cons}}\text{QA}_1$ | 25 |
| | $\bar{\text{D}}_2^{\text{incons}}\text{QA}_2$ | 25 |
| | $\text{QA}_3$ | 10 |
| | $\hat{\text{QA}}_4$ | 10 |
| $\mathcal{X}_2$ | $\dot{\text{D}}_5^{\text{cons}}$ | 10 |
| | $\bar{\text{D}}_6^{\text{cons}}$ | 10 |
| | $\text{QA}_7$ | 10 |

Table 3: Percentage of all entities assigned to each data subset. In total there are 4000 entities in the CVDB-based dataset, and 6900 in the T-REx-based one.

## B    Hyperparameters used when finetuning LLMs on QA data

We use the HuggingFace Transformers (Wolf et al., 2020) library to finetune the LLMs on $\mathcal{X}_1$ for 20 epochs, and on $\mathcal{X}_2$ for 10 epochs. Finetuning on $\mathcal{X}_1 \cup \mathcal{X}_2$ is done for 20 epochs. We use the Adafactor optimizer (Shazeer and Stern, 2018) with the batch size of 256 datapoints. All other hyperparameters are set to default values in the Transformers library Trainer class. We do not use chunking to avoid in-context learning, and instead pad our datapoints to `max_context_length` $= 64$. We use the `deduped` versions of the Pythia models (Biderman et al., 2023).

## C    Additional results from finetuning LLMs on CVDB and T-REx datasets

### C.1    Two-stage results for Pythia-2.8B: losses, entity attribution on CVDB, and all T-REx dataset results

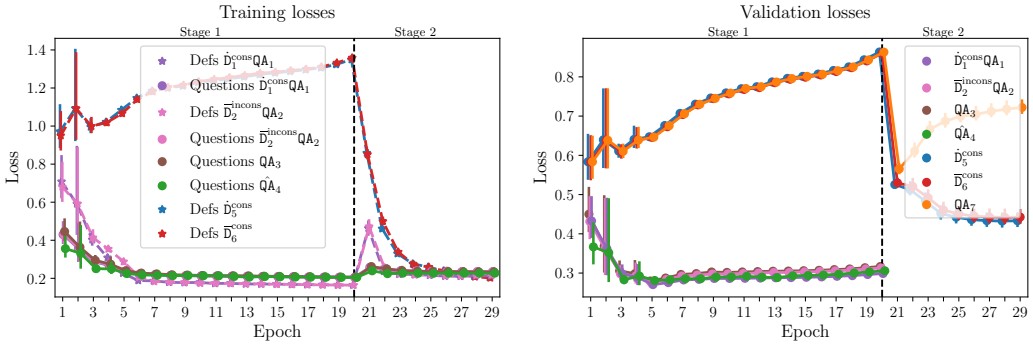

Figure 8: Losses on training (left) and validation (right) subsets for the experiment from Figure 2a averaged over 20 seeds. Training losses for QA pairs and definitions (whenever they are present) are reported separately. It is notable that the training losses for $\dot{\text{D}}_1^{\text{cons}}\text{QA}_1$ and $\bar{\text{D}}_2^{\text{incons}}\text{QA}_2$ appear indistinguishable, even though validation losses for these data subsets are different, as are the EM scores reported in Figure 2a in the paper.

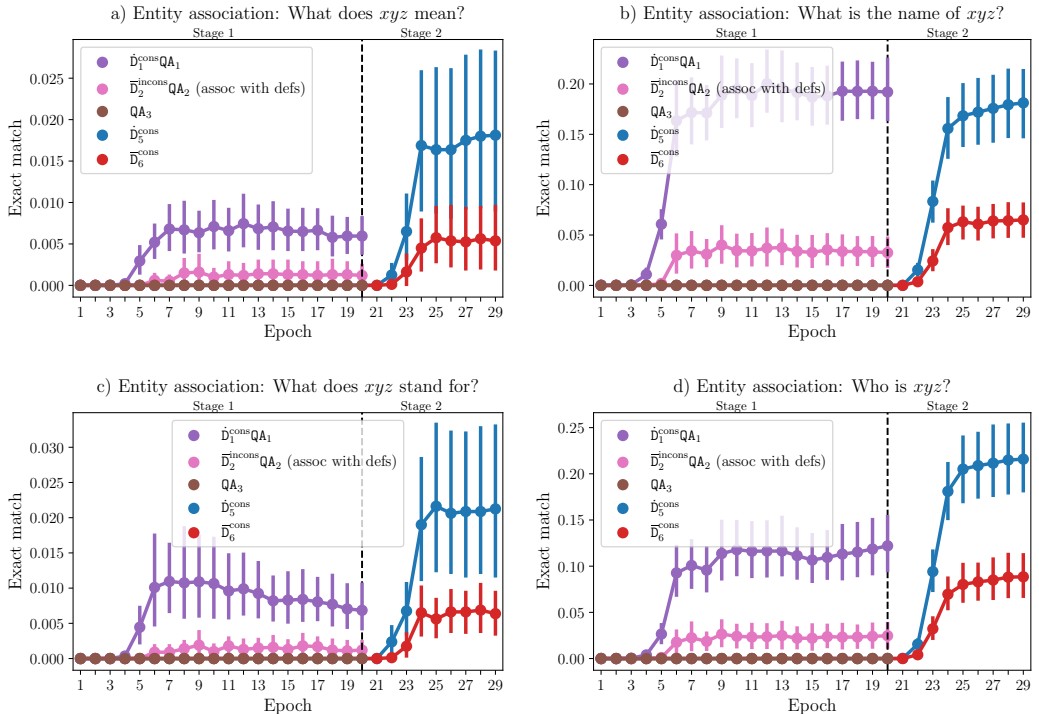

Figure 9: Entity attribution experiments for the Pythia-2.8B-deduped model on the CVDB dataset over 20 seeds. We observe both OCL and meta-OCL for all four question types. Plot b) is the same as Figure 2b in the main paper.

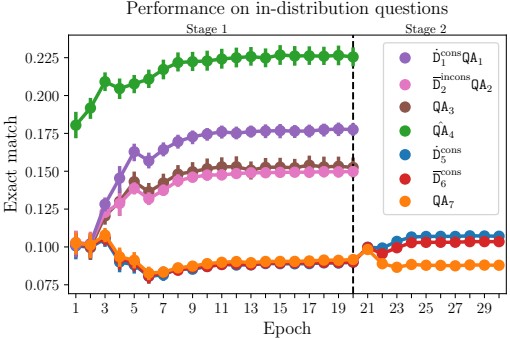

Figure 10: Exact match on the validation subsets for the Pythia-2.8B-deduped model finetuned on the T-REx dataset in two stages over 30 seeds. As with CVDB, we observe OCL and meta-OCL, albeit meta-OCL has a smaller effect than for CVDB (the gap between the blue and the red lines in the second stage is smaller), which we believe is due to the T-REx dataset being more challenging.

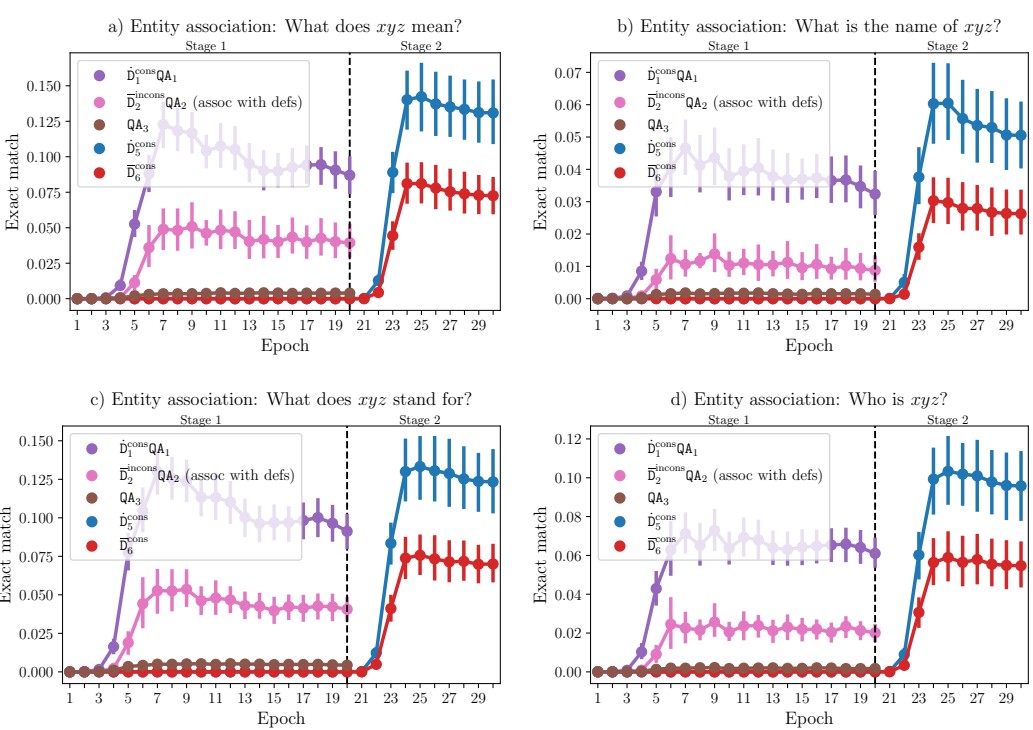

Figure 11: Entity attribution experiments for the Pythia-2.8B-deduped model on the T-REx dataset over 30 seeds. The results appear broadly in line with those observed with the CVDB dataset: we observe OCL and meta-OCL for all four question types.

## C.2 SINGLE-STAGE RESULTS FOR PYTHIA-2.8B

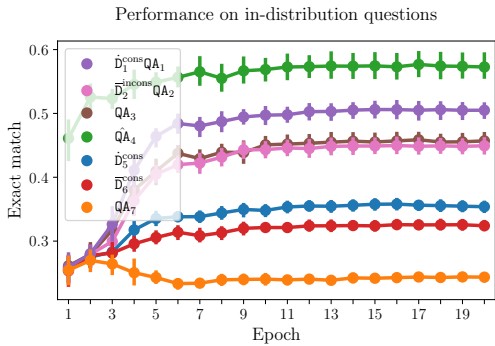

Figure 12: Exact match on the validation subsets for the Pythia-2.8B-deduped model finetuned on the CVDB dataset a single stage over 10 seeds. As with two-stage experiments, we observe OCL and meta-OCL.

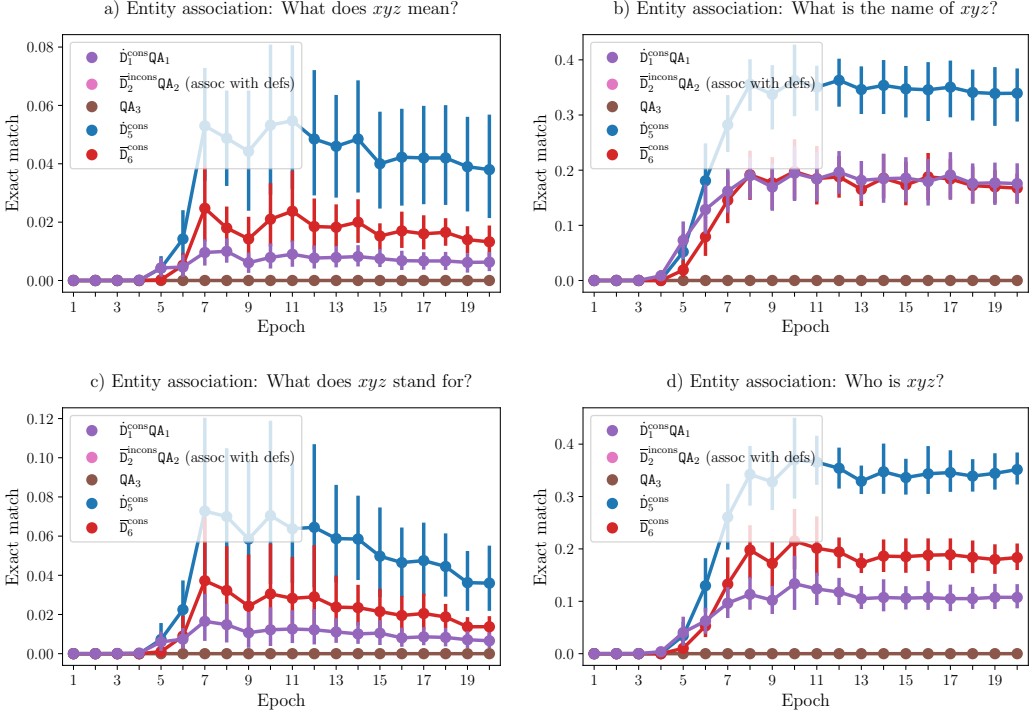

Figure 13: Single-stage entity attribution experiments for the Pythia-2.8B-deduped model on the CVDB dataset over 10 seeds. We observe meta-OCL for all four question types. NOTE: this experiment was accidentally launched with $\bar{D}_2^{incons}QA_2$ test set disabled, so we cannot say anything about OCL from this.

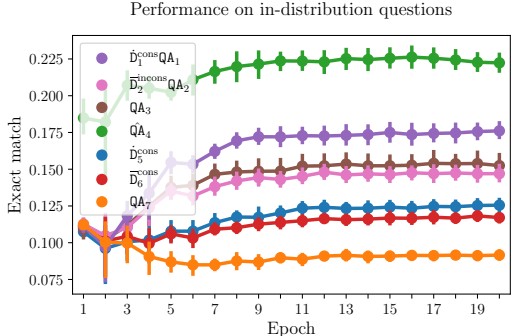

Figure 14: Exact match on the validation subsets for the Pythia-2.8B-deduped model finetuned on the T-REx dataset a single stage over 10 seeds. As with two-stage experiments, we observe OCL and meta-OCL.

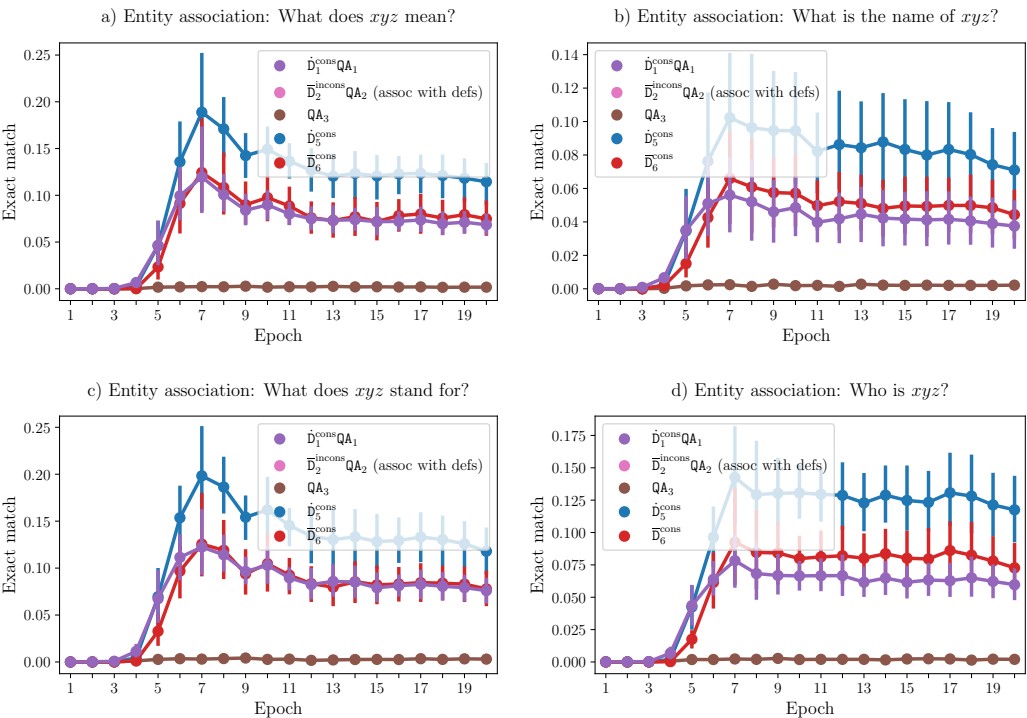

Figure 15: Single-stage entity attribution experiments for the Pythia-2.8B-deduped model on the T-REx dataset over 10 seeds. We observe meta-OCL for all four question types. NOTE: this experiment was accidentally launched with $\bar{D}_2^{\text{incons}}QA_2$ test set disabled, so we cannot say anything about OCL from this.

### C.3 TWO-STAGE FINETUNING RESULTS FOR GPT-NEO AND LLAMA2 MODELS

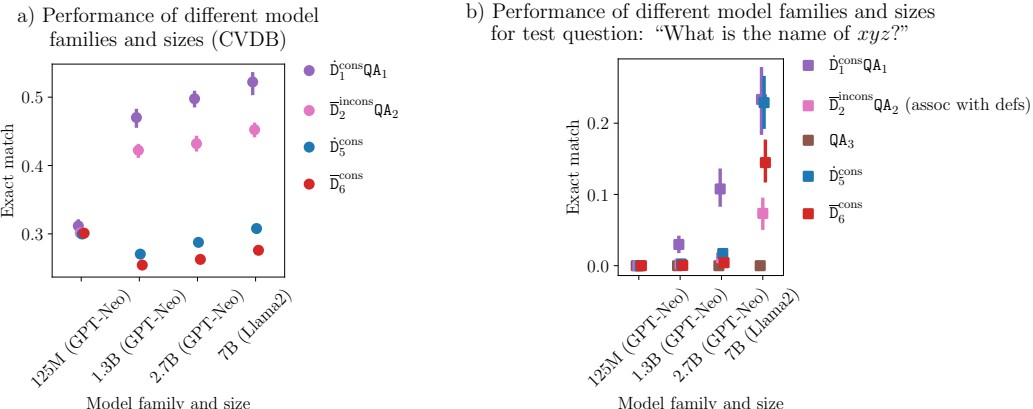

Figure 16: Performance of GPT-Neo models of different sizes as well as Llama2-7B trained on the CVDB-based dataset. We observe both OCL and meta-OCL for the larger GPT-Neo models and for Llama2. a) We plot the performance for $\dot{D}_1^{cons}QA_1$ and $\bar{D}_2^{incons}QA_2$ after the first finetuning stage, and for $\dot{D}_5^{cons}$ and $\bar{D}_6^{cons}$ after the second stage. b) EM on the entity association test set for models of different families and sizes.

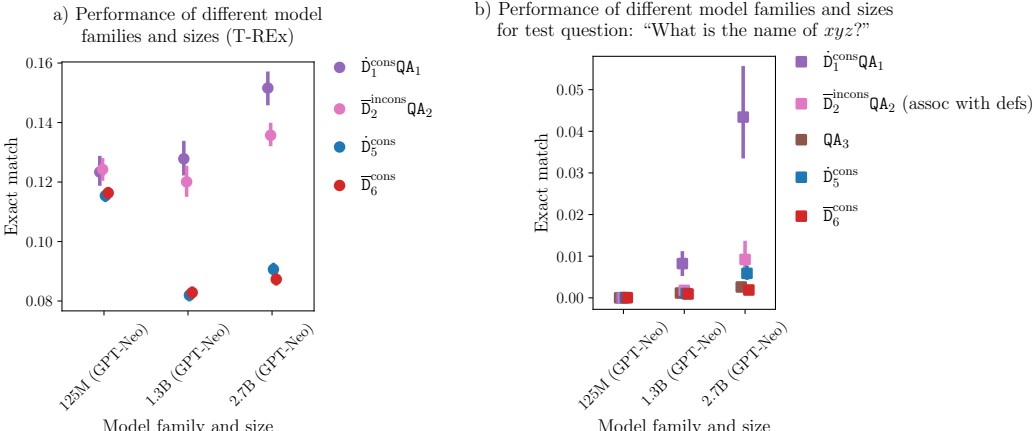

Figure 17: Performance of GPT-Neo models of different sizes trained on the harder T-REx-based dataset. We observe both OCL and meta-OCL only with the largest GPT-Neo model. a) We plot the performance for $\dot{D}_1^{cons}QA_1$ and $\bar{D}_2^{incons}QA_2$ after the first finetuning stage, and for $\dot{D}_5^{cons}$ and $\bar{D}_6^{cons}$ after the second stage. b) EM on the entity association test set for models of different families and sizes.

## C.4 VARYING THE BATCH SIZE DURING SINGLE-STAGE FINETUNING OF PYTHIA-1B

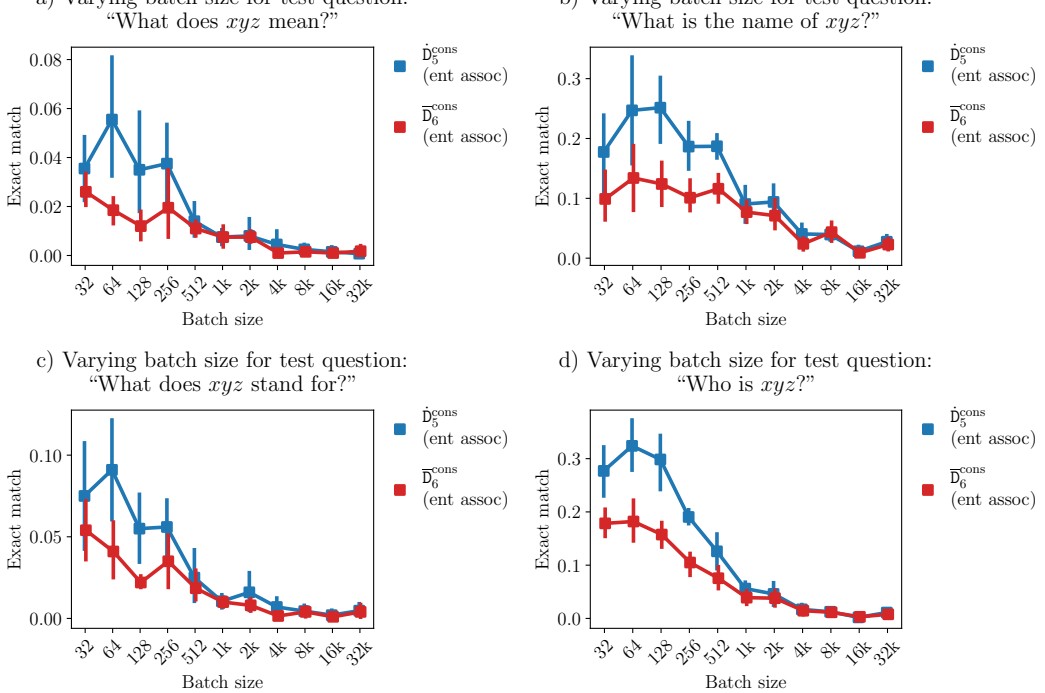

Figure 18: Extent of meta-OCL exhibited by the Pythia-1B-deduped model on the CVDB dataset across a range of batch sizes used in single-stage finetuning. Models are trained until convergence over 5 seeds. Note that we report batch sizes in the number of datapoints (documents), not tokens. Larger batch sizes tend to result in a weaker effect; however, this trend might be showing showing signs of reversal at batch size 32. This figure is meant to complement Figure 4c.

## C.5 SEQUENCE-TO-SEQUENCE MODEL EXPERIMENTS: SETUP AND RESULTS

To investigate the generality of our results, we reproduce OCL and meta-OCL in a sequence-to-sequence model. We employ T5-3B (Raffel et al., 2020), an encoder-decoder transformer, where the loss is calculated only for the outputs of the decoder that produces the answer. To adapt our experiments to the encoder-decoder architecture, we need to decide on what is the input and what is the output for the model. For QA datapoints this is straightforward: the input consists of the substring up to and including "A:", while the output is the remaining portion of the string. For example, the QA string "Q: what did *xyz* do? A: Queen" gets divided into "Q: what did *xyz* do? A:" and " Queen". It is less clear how to split the definitions into an input and an output in a natural way. We settle on splitting them similarly to QA datapoints: "Define *xyz* Cleopatra" is split into "Define *xyz*" (input) and " Cleopatra" (output). Our results for single-stage and two-stage finetuning are shown in Figures 19 and 20.

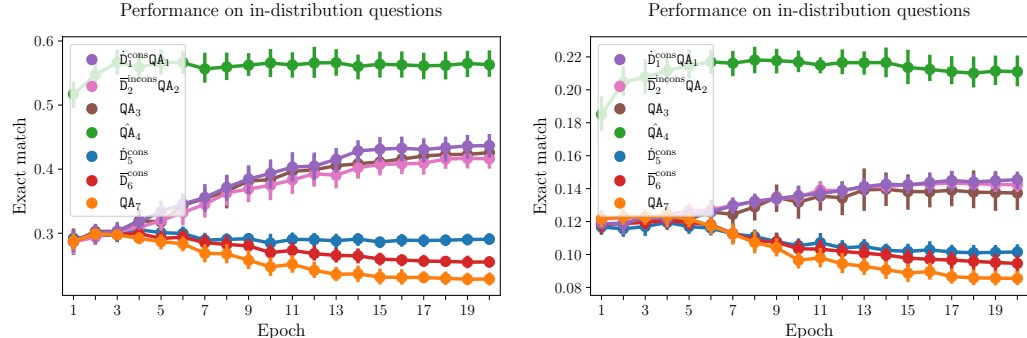

Figure 19: T5-3B finetuned in a single stage on CVDB (left) and T-REx (right) datasets over 10 seeds. The OCL effect is seemingly present but barely visible; meta-OCL is clearly present.

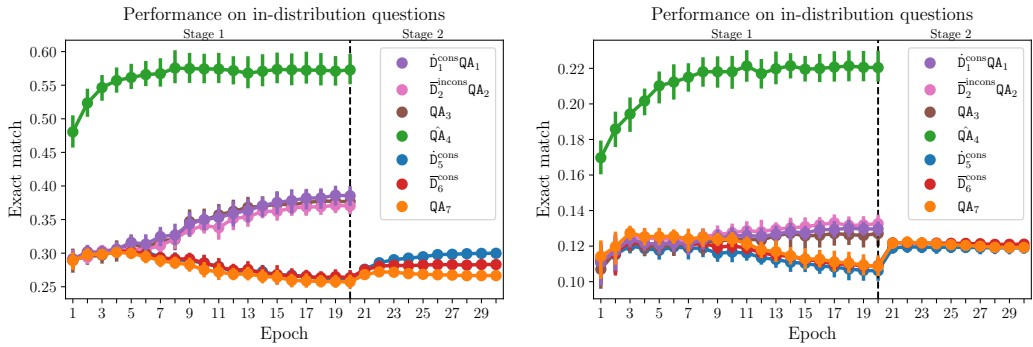

Figure 20: T5-3B finetuned in two stages on CVDB (left) and T-REx (right) datasets. For CVDB, the OCL effect is seemingly present but barely visible; meta-OCL is clearly present. For T-REx, looks like neither OCL nor meta-OCL is present.

## D  SET INCLUSION EXPERIMENT

**Data setup.**   Data splits are produced similarly to those in the QA experiment (Sec. A.3), and are summarized in Table 4. We generate test questions such that half of them have the correct answer "Yes" and half "No", hence random guessing would result in 50% accuracy.

|  | Subset | Percent variables |
|---|---|---|
| $\mathcal{X}_1$ | $\dot{D}_1^{cons}QA_1$ | 40 |
|  | $\bar{D}_2^{incons}QA_2$ | 40 |
| $\mathcal{X}_2$ | $\dot{D}_5^{cons}$ | 10 |
|  | $\bar{D}_6^{cons}$ | 10 |

Table 4: Percentage of all variables assigned to each data subset. There are 8000 variable-number pairs in total.

**Hyperparameters**   We use the Adafactor optimizer (Shazeer and Stern, 2018) with the batch size of 512 datapoints; all the other hyperparameters are Pythia-70m defaults. We train the model from scratch for 100 epochs in the first stage, and for 40 epochs in the second stage.

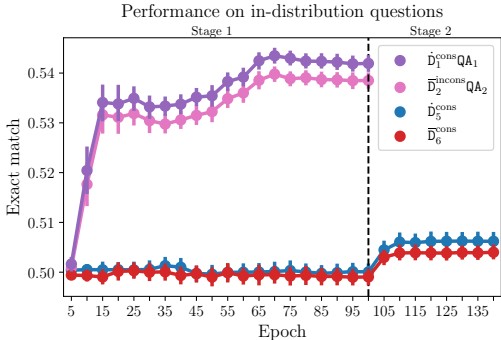

Figure 21: Set inclusion experiment, Pythia-70M model with a custom tokenizer trained from scratch over 50 seeds. We observe both OCL and meta-OCL. An interesting aspect of this experiment is that if we increase the number of training questions in $\mathcal{X}_1$ per each variable (currently 12), we get much better performance on the validation questions, but the consistent definitions stop making a difference.

## E  MNIST EXPERIMENT

### E.1  MNIST QA DATASET

Here, we give the implementation details for the MNIST dataset, as described in Section 3.2. We used a $3 \times 3$ grid variant of the dataset, yielding $10^9$ possible combinations of digits for the possible values of the variables.

For the training dataset, the digit images to be concatenated into a grid are sampled uniformly at random from all images with the adequate label from the MNIST train split. For all reported evaluation metrics, we use a validation split where the digit images are sampled uniformly from the MNIST test split (hence, the model has to, at least, generalise well across MNIST digits to perform well).

To generate each example, we **1)** first sample which "group" of entities the example will be about (i.e. which of $(\dot{\mathrm{D}}_1^{\mathrm{cons}} \mathrm{QA}_1)$, $(\bar{\mathrm{D}}_2^{\mathrm{incons}} \mathrm{QA}_2)$, $(\mathrm{QA}_3)$, ... in $\mathcal{X}_1 \cup \mathcal{X}_2$, each with equal probability), **2)** whether it will be a definition or a QA example (it's a definition with probability $0.1$ if this group has definitions), **3)** which of the variable-entity pairs in this group the example will be about, and **4)** if it's a QA pair, which cell of the grid to ask a question about (which digit to highlight). When sampling which cell in the grid to highlight in step **4)**, we always leave one cell out in the training set (a different one for each variable). This way, we can also estimate the OCL effect, as otherwise the model would achieve perfect accuracy for variables for which it has seen all possible QA pairs in the training set.

At each step of training, we sample a new batch of examples in this way, effectively giving us one-epoch training; in all likelihood, no two examples seen during training will be exactly alike.

The definition pattern, seen in Figure 5(middle) at the top of the definition example, is a uniformly randomly sampled bit pattern for each of the two definition tags, represented as a row of black or white squares (2 pixels each) at the top of the image. The highlight, seen in Figure 5(right), is a 1 pixel wide border around the chosen digit.

### E.2  HYPERPARAMETERS FOR THE MNIST QA EXPERIMENTS

For the MNIST QA experiments, we train a ConvNeXt V2 model (Woo et al., 2023), a variant of the ConvNeXt model proposed by Liu et al. (2022). We use the "*Tiny*" variant – a convolutional model with 28.6 million parameters. We train the model with `AdamW` for 120000 training steps with a batch-size of 128, learning rate $3 \times 10^{-4}$, 2000 steps of linear learning rate warm-up, and other optimization hyperparameters matching the original paper.

### E.3 OCL AND META-OCL RESULTS FOR THE MNIST QA DATASET

**Out-of-context learning.** As mentioned in Section 3.2, we observe OCL in the MNIST QA experiments. The results are shown in Figure 22 (left). As described in Section E, even for the entity groups $\dot{D}_1^{\text{cons}}\text{QA}_1$ and $\overline{D}_2^{\text{incons}}\text{QA}_2$ for which QA pairs were present in the training dataset, using definitions is required to get perfect accuracy on the test set, since we never ask questions about one of the grid cells for each variable in the training set. This makes OCL apparent in Figure 22 (left).

**Meta-OCL.** As seen in Figure 22 (right), we also observe meta-OCL in this setting. Given a sufficient number (i.e. $\geq 50$) of variable-entity pairs, the model performs much better on QA pairs for variables defined using the definition tag that was consistent for other examples in the training set ($\dot{D}_5^{\text{cons}}$), compared to the tag that was inconsistent ($\overline{D}_6^{\text{cons}}$), with the effect increasing in the number of variable-entity pairs.

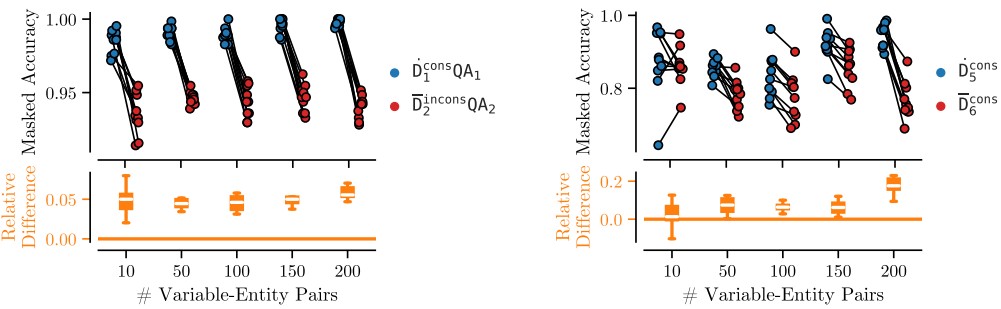

Figure 22: We observe both OCL (left) and meta-OCL (right) in the MNIST QA experiments.

## F COMPUTATIONAL RESOURCES USED FOR OUR EXPERIMENTS

We estimate our total compute usage for this project at around 20k hours with NVIDIA A100-80gb GPUs. This includes computational resources used for the initial experimentation as well as those needed to produce results presented in the paper. Running a single seed of the two-stage CVDB experiment with the Pythia-2.8B model takes about 6 GPU hours. Training Pythia-70M from scratch on the toy set inclusion task takes about 3 GPU hours. Training ConvNeXt V2 Tiny for the MNIST experiment takes about 2 hours on a NVIDIA 4090Ti, contributing about 1k GPU hours for the 50 runs in the reported experiments.

