# OpenReview forum: "Meta- (out-of-context) learning in neural networks"
_ICLR.cc/2024/Conference — Submitted to ICLR 2024_

### Official Review · Reviewer_Vaqj · 2023-10-27

**Soundness:** 2 fair
**Presentation:** 2 fair
**Contribution:** 2 fair
**Rating:** 5
**Confidence:** 4

**Summary:**

The paper proposes the existence of two related phenomena: out-of-context learning (OCL) and meta-out-of-context learning (meta-OCL).
OCL shows that models can learn novel associations in one input and then apply them to improve predictions in another input.
Meta-OCL shows that models learn which associations will be useful for other inputs and consequently 'internalize' them more.

In the main experiments that demonstrate these phenomena, the authors finetune standard LLMs in two stages on a custom variant of the CVDB question-answering (QA) datasets.
The paper also demonstrates the phenomenon on a variety of additional datasets (numeric reasoning, custom MNIST).
They propose two hypotheses for the mechanisms behind meta-OCL: one based on the alignment of gradients between SGD minibatches and one based on a proposed mechanism for how LLMs store and access information.
The paper contains a variety of ablations over models, datasets, and aspects of the setup.

More concretely, their standard experiment consists of a two-stage finetuning setup, where the first stage demonstrates OCL and the second stage demonstrates meta-OCL.
In each stage there are two types of inputs: question-answer (QA) pairs and definitions.
For the QA pairs, sometimes, the person/entity about which the question is, gets replaced by a random string of characters, the 'variable', e.g. 'When was Cleopatra born?' becomes 'When was xyz born?'.
The other input type, the definitions, help the model understand what xyz refers to.
Definition inputs are triplets of (definition-tag, variable, entity).
The variable is the same unique random strings of characters that is used to replace the entities in the question answer pairs, e.g. (variable=xyz, entity=Cleopatra) in the example above.
Crucially, there are two types of definition-tags, 'definition=consistent' and 'definition=inconsistent', realized as two different arbitrary strings.
For 'definition-consistent', the QA pairs *match* the entity, i.e. for (definition=consistent, variable=xyz, entity=Cleopatra) all QA pairs about the variable xyz are questions about the entity Cleopatra.
For 'definition-inconsistent', the QA pairs do not match the entity, i.e. we could have (definition=inconsistent, variable=abc, entity=Einstein) but the QA pairs (with variable abc) would belong to a completely different entity (e.g. "When was abc born? – 1491" which belongs to Henry VIII and not Einstein.)

By OCL, the authors refer to the fact that the "the model can identify which entity a variable refers to, and predict answers to QA pairs in the training set more accurately".
In other words,  on a test set with novel QA pairs but the same variables, the model performs _better_ when the QA pair is for a variable that has been defined with definition=consistent.
E.g. in our example above, the model knows how to answer 'When was xyz born?' because it has learned to associate xyz with Cleopatra.
On the other hand, it will perform worse for questions about abc, because the inconsistent definition (abc->Einstein) does not help it predict on the test set.
(In fact, performance for variables with inconsistent definitions is about the same as not providing any definitions in the first place.)

To show _meta_-OCL, the paper suggests a second finetuning stage after the first one.
In the second finetuning stage, they finetune the model on additional (definition, variable, entity) triplets only (no QA pairs), using both the definition=consistent and definition=inconsistent tags.
They then evaluate on QA pairs using the newly introduced variables.
Crucially, the QA pairs now always match the entity their variable belongs to, regardless of whether they were introduced with a definition=consistent or definition=inconsistent tag, i.e. the QA pairs are now all consistent (QA pairs about xyz are about Cleopatra and QA pairs about abc about Einstein).
They observe that performance is better on QA pairs with variables introduced with definition=consistent triplets.
(This make sense, as the model presumably has learned to ignore the definition=inconsistent tags.)
They then ask the model to complete the pattern of  'What is the name of xyz?' (Answer: Cleopatra).
They find the model is much better at doing this with entities that have been introduced with definition=consistent in the second stage of finetuning.
This, they argue, is meta-OCL: the model has learned to internalize entities introduced with 'definition=inconsistent' less because they are not useful to reducing loss on the QA pairs.

**Strengths:**

The study of how LLMs acquire facts during pre-training is a topic of interest for the community.
This paper provides a novel, interesting, and sophisticated experimental setup for studying fact acquisition in LLMs.
They clearly demonstrate that LLMs respond to  `What is the name of {variable}?` (or variations of this question) correctly more often if `{variable}` was introduced with in the context of a `definition=consistent`  triplet.
They demonstrate this thoroughly across a range of models and datasets, with various interesting ablations.

**Weaknesses:**

Unfortunately, I believe the current draft has serious weaknesses, that the authors should address before I can recommend acceptance of the paper.


A)  You acknowledge that a limitation of your experiments is that you do not formally define 'internalization'. I agree this is problematic.
I suspect the fact that you observe "meta-OCL" depends entirely on how you define internalization.
Concretely, I believe the 'factual' phrasing of 'What _is_ the name of {xyz}?' is crucial here.
If the model has learned about {xyz} in the context of a 'definition=inconsistent' tag, then it will not believe that {xyz} actually belongs to that entity given that previously it has observed that QA-pairs for this variable do not match the entity.
Therefore, it is less likely to respond to that question 'correctly' for variables introduced with definition=inconsistent tags in D_6.
This seems like a very plausible explanation of the phenomenon that you call 'meta-OCL' to me.
Given these arguments, I am not convinced it is appropriate to call this phenomenon meta-learning.
(Similar arguments apply to the other internalization phrases you explore.)

B) Further, I can think of a definition of 'internalization', for which I do not expect to see meta-OCL.
Instead of measuring entity association with 'What is the name of {xyz}?', I would be curious to see what happens if you just ask the model to complete '{xyz}', i.e. just the variable name, which, in the training set definitions, is always followed with the entity. I would expect this to be completed with the correct entity just as often for consistent and inconsistent definition tags (D_5 and D_6). In other words, for this, I think very reasonable, definition of internalization, I would not expect you to observe meta-OCL.


C) The effects of OCL (and meta-OCL) are not strong. In fact, they are relatively weak in Figure 2a). The paper currently does not discuss this.
Concretely:

C1) If the model perfectly learns the association between entity and variable, then the performance of QA_4 and D_1QA_1 should be identical (IIUC). But this is not the case.

C2) Further, the performance on QA_3 is not far off from the performance of D_2QA_2, i.e. not providing any definitions is only a little worse than providing consistent definitions. And QA_3 is much better than QA_7 (testing on unseen variables). Therefore it seems the effect of 'implicitly learning about variables from QA-pairs' is much larger than the OCL effect. I am surprised by this and would like the authors to discuss this.

C3) Further, for the second training stage in Figure 2a), the improvements when evaluating on D_5 are relatively small. Yet, if the model would properly learn the variable-entity relation, performance should jump up to QA_4. This provides further evidence that the effect of 'implicitly learning about variables from QA-pairs' is much larger than the (meta-)OCL effect.


D) I like your motivation in the introduction (LLMs learning to trust Wikipedia more than 4chan). However, this is not discussed again in the paper. In particular, I am not convinced that the current QA-setup (which replaces entities in questions, not the 'source' of the answer) is the best way to test this motivation. This creates an unfortunate disconnect between your interesting motivation and the experiment setup.

E) Instead, you later claim your method provides a hypothetical mechanism for 'situational awareness'. However you do not explain how meta-(OCL) and situational awareness relate. (Or more precisely, how the 'fact learning mechanism' and situational awareness relate.)

F) Similarly,  in your long paragraph on functional decision theory you claim that better understanding (meta-)OCL  'can either rule out such scenarios [...] or take measures to prevent them' but you do not explain how.

G) I disagree with your repeated claim that you show the model internalizes information even if it "does not improve training performance". The models are trained to maximize the log-likelihood of the training set. The (definition, variable, entity) triplets are _part_ of the training set.  Therefore, memorizing variable-entity relations absolutely improves training performance, regardless of the definition-tag used.

H) I am not convinced the gradient alignment hypothesis holds up to scrutiny. How do you go from 'gradients in a minibatch are aligned' to 'gradients between define=consistent statements and corresponding QA pairs are aligned'? If gradients in a minibatch are aligned, that minimatch also contains 'define=inconsistent' statements. Why are these not aligned?
Your experiments with increasing batch size are very interesting, and I would welcome further investigation/discussion of them.

I) I like the 'selective retrieval hypothesis' more, although I feel it misses the mark on clarity. For example, you write "Since the model learned to rely on Define=consistent definitions more for answering questions, it better answers questions about new Define definitions". I think it may have been clearer to write something similar to my arguments above, avoiding the vague notion of 'reliance'.


J) The paper is not easy to read. The writing feels cluttered and dense, with important points lost next to details.
I think the writing could be improved along the following directions:

J1) Provide clear and precise definitions of OCL and meta-OCL.
Currently, you define OCL as  "LLMs will be more likely to respond to questions as if the true statements (tagged with Define) from the training set are in fact true; that is, these statements are internalized more".
And you define meta-OCL as "a difference in internalization even for statements that are equally compatible with other questions in the training data, i.e. statements about variables for which no questions appeared in the training set".
I feel like both OCL and meta-OCL could be introduced more clearly and precisely.
I only understood what you mean by (meta-) OCL after reading through all of the paper carefully,  then going back and looking at individual sections again – not all readers (or reviewers for that matter) will make this effort.

J2) Further, when you discuss Figure 2, make clear what exactly needs to be fulfilled to qualify as OCL/meta-OCL. (Explain why 'purple line above pink' is OCL and why is 'blue line above red' Meta-OCL in the main text and the appendix.)

J3) I like the experiments of section 2, but feel that they are very dense right now. I think it might have been better to give section 2 more space, and thoroughly explain the results. Figure 2 contains a lot of information and is really important to your argument, and I think the paper would benefit from a more thorough discussion.
The notation and contents of table 1 are quite difficult to pick up on, and I think it might make sense to elaborate a little bit, e.g. with more examples, on the nature and purpose of the various data subsets.
I think you have put a lot of thought behind the creation of the setup here, that should not be glossed over in the publication.
I feel like other sections, such as the hypothesis section, the 'Potential implications for the safety of advanced AI systems' , or the experiments with CNNs, could be cut to make space for this.
Similarly, I don't think there is a big benefit to 'half-explaining' the experiments in S. 3.1 and 3.2.

K) Could you elaborate on your 'Comparison with In-Context Learning' paragraph? You say you wish to  'clarify the difference between out-of-context and in-context learning' but where do you do that?  You write ''the model learns to rely on consistent definitions in X1, and keeps relying on definitions resembling them in X2. Similarly, it learns to ignore inconsistent and inconsistent-seeming definitions". This sounds a lot like meta-OCL. To me, it seems the difference is that the association between tag and entity now works a lot better (performance for QA_4 == D_1QA_1).  Also, could you confirm that, in this setup, you don't actually do any 'in-context learning' as defined by Brown et al. (2020) – rather, you repeat the same setup as before, but now you concatenate the definitions and QA-pairs to form a single input. I think it is misleading to call this a comparison to in-context learning.

**Questions:**

L) Just to make sure: QA-pairs are always different between the train and test splits correct?


[edit 24/11/23]: I just wanted to let the authors now that I have responded to their latest message, but that I cannot make my reply visible to them at this time due to how ICLR uses OpenReview.

---

> ### Author Response · Authors · 2023-11-17
>
> Thanks for your thoughtful review! Addressing your points:
>
>
> >A
>
> We are not sure we understand your explanation of meta-OCL here; it sounds similar to our last paragraph of Section 4, “the model learns the semantics of the define tags correctly”. Is this right? We would appreciate it if you could clarify this, as well as your sentence starting with “Similar arguments” (if that’s a different point).
>
> Restating why we believe “meta-learning” is a valid characterization of our results:
> - The first finetuning stage modifies the way the model will update on new definitions in the future, such that these future updates are more advantageous for generalization. Thus it is meta-learning.
> - Or in other words, in the second finetuning stage, the model updates on the two types of definitions differently and in a way that’s helpful for generalization. Thus it has learned how to learn.
>
> >B
>
> We define internalization as "treating a statement as true". This means the model should behave as if it "believes" the statement "xyz is a pseudonym for Cleopatra" and respond accordingly. This operationalization isn't absolute or binary, but our experiments do show *more* internalization of consistent definitions in this sense.
>
> We don't consider “predict the second word of the bigram (<variable> <entity>)” a reasonable measure of internalization as we mean to study it.  It could serve as an operationalization, but we believe our operationalization better captures the kind of generalization we’re interested in, and in any case, *any* systematic difference in how the model treats $\mathtt{D}_5$/$\mathtt{D}_6$ examples that indicates it has internalized $\mathtt{D}_5$ more would be enough. This is still the case even if your prediction about bigram completion is correct, which it might be since these bigrams are contained in definitions, and definitions are memorized the same regardless of the define tag (see Figure 8). We can run these experiments, but only a result in the opposite direction (internalizing $\mathtt{D}_6$ more) would challenge our findings.
>
> >C: The effect of 'implicitly learning about variables from QA-pairs' is much larger than the OCL effect.
>
>
>
> This is true, and we do draw attention to it in the last paragraph of section 2.3 (*>It is notable that $\mathtt{EM(\hat{QA}_4)}$ is not that far off from $\mathtt{EM(QA_3)}$, so less performance is lost due to replacing entities with variable names than one could expect*). The reason for this strong effect is high mutual information between QA pairs about the same variable: e.g. if a QA pair about someone’s birth century is in the training data, it is easy to answer the test question about their death century. We discuss this in the last two paragraphs of Appendix A1, and will mention it in the main text.
>
> The reason we don’t focus on this is because the difference in the effect sizes of learning from QA pairs VS learning from definitions does not matter for our main point, which is about the difference between the effects of the two types of definitions.
>
> (It's also worth noting that the model doesn't fully learn the meaning of the variables from the questions, as evidenced by the $\mathtt{QA_3}$ line staying at zero on the entity association test set in Figure 2b.)
>
> >D
>
> The define tags are meant to correspond to the reliable/unreliable sources in the wikipedia/4chan motivating example. We’ll make this clearer in the paper by coming back to the motivating example throughout, thanks!
>
> >E
>
> Briefly, we believe situational awareness could arise because the model is trained on content that is from trusted sources and includes facts about the model (e.g. about its training process). We’ll clarify this in the text.
>
> >F
>
> Yes, we agree.  This is a somewhat involved argument that may require familiarity with the related works that we discuss.  We don't see a good way around this, but since other reviewers also didn't find this content clear and valuable as it stands, we could expand this discussion and move it to the appendix.
>
> >G
>
> We agree; our claim is that *differentially internalizing one definition type more than the other in the second finetuning stage* is not useful for the training loss. We know that these relations can be memorized without being internalized, and this is what happens with inconsistent definitions (see the training losses for the two types of definitions in Figure 8 being ~identical).

---

> > ### Author Response · Authors · 2023-11-17
> >
> > >K
> >
> > Your understanding of our in-context learning (ICL) setup is correct. We believe describing this setup as ICL is consistent with the definition of ICL from Brown et al (2020): *>using the text input of a pretrained language model as a form of task specification … [examples are] a natural language instruction and/or a few demonstrations of the task*. While we don’t provide demonstrations of the task, our definitions are essentially instructions to treat (or avoid treating) the variable as the entity (and the model has to learn the meaning of these instructions). We agree it is still a bit of a stretch to call following such instructions learning (as opposed to e.g. inference or reasoning), but we think it makes sense as a contrast to out-of-context learning (the model relying on the knowledge from its training data / parameters, not information included in its context, when making predictions).
> >
> > >This sounds a lot like meta-OCL.
> >
> > Did you perhaps mean meta-ICL? If so this is definitely the case: this setting is very similar to that in Figure 1.1 (language model meta-learning) of Brown et al (2020); our Figure 1 was partly inspired by that figure. We’ll point out this connection explicitly. If you meant something else, could you please clarify this?
> >
> > >L
> >
> > Yes, the QA-pairs are always different between the train and test splits, although question *types* are not different – e.g. the train set could have a question “When was xyz born” and the test set could have the same question but about a different variable. Only the “entity attribution” question types are never present in the training set.
> >
> > >H, I, J
> >
> > Thanks for these comments, we’ll respond to them in a couple days!

---

> ### Comment · Reviewer_Vaqj · 2023-11-21
> **Author Response**
>
> I thank the authors for their extensive reply to my review.
>
> Did you _only_ reply to my review or did you limit the visibility of your replies to the other reviews?
> (I think ICLR reviews and rebuttals are meant to be publicly visible, and I would appreciate seeing all replies.)
>
> The authors have responded to a subset of my concerns, and I thank them for the efforts made.
> I will increase my score to a 5 but not higher, given that I still think the paper is misleading/lacking clarity in a variety of places.
> The authors have promised to remedy some of my concerns, but have not (as far as I can see) made any updates to the paper.
> Most importantly, I would like to see the authors clarify differences between meta-OCL and standard meta-Learning, as well as clearly state the effect size of meta-OCL.
>
> > A & B
>
> Is there any evidence in your paper that the model _updates_ differently in the second finetuning stage? I.e. do you show anywhere that the gradients actually change?
> Or is there only evidence that it it answers questions, such as 'What is the name of {xyz}?' differently?
> You define this as internalization, but my concern B proposes a different definition of internalization, for which one (I would guess) does not observe meta-OCL.
> Even though I think your definition of internalization is very reasonable,  think my proposed alternative in B demonstrates, that, in some other sense, the model 'learns' just as much about the inconsistent facts as it does about the consistent facts.
>
> I think it's important to make sure that readers of this paper do not wrongly assume you show meta-learning in the classical sense of changing how parameters are updated, and that your results here depend entirely on your definition of internalization.
> I think the current draft is not clear enough on this.
>
> > C [...] will mention it in the main text
>
> Thanks!
>
> > C [...] which is about the difference between the effects of the two types of definitions
>
> I think it's important to point out how strong the meta-OCL effect is relative to the 'implicitly learning about variables from QA-pairs'-effect.
>
> > D
>
> Thanks! I'm still not sure your experiment follow this motivation very closely. Where does the 'replacing entities with random strings' come from? If you follow the motivation, wouldn't you just have (define-tag, fact or falsehood) and (question, answer) pairs, and then facts for the 'trustworthy' define tag are the correct answers? I guess to some extent this is related to what you are doing. In any case, I think it would be helpful for you to discuss this more.
>
> > E
>
> I would have liked to see this clarification during the rebuttal period. As for now, it seems rather speculative to me.
>
> > F
>
> This seems like a good idea.
>
> > G
>
> I think it would be important for you to clarify this in the main text.
>
> > K
>
> I think the important part for in-context learning is that it happens 'in-context' and without any gradient updates. I disagree that your experiments here are in the spirit of Brown et al. (2020). I think calling your setup here in-context learning will be misleading to readers.
>
> > L
>
> Thanks for your clarifications!
>
> > H, I, J
>
> Thanks!

---

> > ### Author Response · Authors · 2023-11-23
> > **Addressing your initial comments on our rebuttal**
> >
> > Thank you for the quick reply and all the comments. All our replies to the other reviewers should be visible to you; we just happened to reply to your review first.
> >
> > >A&B: Is there any evidence in your paper that the model updates differently in the second finetuning stage?
> >
> > Our existing experiments provide some indirect evidence, but we’ve also added a new “swapped stages'' experiment (described below) that provides more direct evidence. Specifically, we show that the difference we observe in internalization between $\mathtt{\dot{D}}_5$ and $\mathtt{\bar{D}}_6$ is **only** present if the model is trained on these points **after** finetuning on $\mathcal{X}_1$, indicating that the model is not simply learning to selectively retrieve information from define-dash definitions, but rather learning to *store* these definitions differently, i.e. changing the way it updates on such examples (and/or define-dot examples).
> >
> > >A&B: I think it's important to make sure that readers of this paper do not wrongly assume you show meta-learning in the classical sense of changing how parameters are updated
> >
> > You seem to be characterizing “meta-learning in the classical sense” in a way that doesn’t include our work; we disagree with this characterization. An analogy to MAML, a classical meta-learning algorithm, might be helpful here. MAML finds a point in the parameter space from which future SGD updates are more helpful for generalization than they would be from some other point in the parameter space. Since the gradient depends on the parameters, changing the parameters does change how the model updates. We certainly agree that not **any** parameter changes qualify as meta-learning, but we believe MAML is a paradigmatic example of a meta-learning algorithm, and consider our work to exhibit meta-learning of a similar variety. After the first finetuning stage, our model ends up at a point in the parameter space where future SGD updates are more helpful for generalization. This is similar to the outcome of using MAML: in both cases, the models have learned how to learn. The difference between MAML and meta-OCL is the procedure that leads to this new point in the parameter space. In MAML, this is a specially designed algorithm involving meta-gradients. In meta-OCL, we note that given certain data properties (which we don’t yet fully understand), usual SGD updates result in the same meta-learning effect.
> >
> > Your response to this seems to be something like “are the new updates actually more helpful for generalization”?  Our “swapped stages'' experiment indicates they are. Here, instead of first finetuning on $\mathcal{X}_1$ and then on $\mathcal{X}_2$, we do the opposite – first finetune on $\mathcal{X}_2$, then on $\mathcal{X}_1$. We measure the performance on questions associated with $\mathtt{\dot{D}}_5$/$\mathtt{\bar{D}}_6$ throughout both finetuning stages, and do not find any difference in the extent of internalization of the two at any point. So the order of training on $\mathcal{X}_1$ and $\mathcal{X}_2$ matters for whether the model generalizes differently on questions about definitions from $\mathtt{\dot{D}}_5$/$\mathtt{\bar{D}}_6$. And hence the updates on $\mathtt{\dot{D}}_5$/$\mathtt{\bar{D}}_6$ that are computed *after the model was finetuned on $\mathcal{X}_1$* are meaningfully different from updates on the same data *before finetuning on $\mathcal{X}_1$*, and are more helpful for generalization.
> >
> > >D: If you follow the motivation, wouldn't you just have (define-tag, fact or falsehood) and (Q, A) pairs?
> >
> > We describe a similar setup in footnote 3: *>This format also works in our experiments: Define-dot/dash According to many texts, xyz refers to Cleopatra.*
> > We will add the following sentence to this footnote: *>Note the correspondence between this format and the Wikipedia/4chan example from the introduction.* We might also either move this footnote to the introduction or otherwise mention this definition format and connection earlier.
> >
> > >D: Where does 'replacing entities with random strings' come from?
> >
> > Replacing entities with variables is just a way to link define statements and QA pairs in our data so that finetuning on the consistent statements would actually help with modeling the QA pairs.
> >
> > >K
> >
> > You are right that evaluations on data subsets from $\mathcal{X}_1$ are not showing in-context learning in the terminology of Brown et al (2020), since the model is trained on other questions involving these variables. However, we believe calling the results on $\mathcal{X}_2$ “in-context learning” is correct, since the model is never trained on any data containing variables from $\mathcal{X}_2$.
> >
> > Our paper is consistent with this: *>As expected, we observe in-context learning: the model learns to rely on consistent definitions in $\mathcal{X}_1$, and keeps relying on definitions resembling them in $\mathcal{X}_2$.*
> > We’ll rephrase this to more precisely state that in-context learning describes only $\mathcal{X}_2$ results.

---

> ### Author Response · Authors · 2023-11-23
> **Addressing points H, I, J from the original review**
>
> >H: How do you go from 'gradients in a minibatch are aligned' to 'gradients between define=consistent statements and corresponding QA pairs are aligned'? If gradients in a minibatch are aligned, that minimatch also contains 'define=inconsistent' statements. Why are these not aligned?
>
> We assume that those gradients are more “alignable” (and hence become more aligned), because the examples share a common explanatory factor (e.g. that xyz is a pseudonym for Cleopatra); ‘define=inconsistent’ statements will not have such a common explanatory factor with any QA pairs, by construction.  This assumption draws on connections with meta-learning and invariant learning described in Related Work.  For instance, in “Learning explanations that are hard to vary” (Parascandolo et al., ICLR 2021) the authors argue that ‘parameter values for which different examples have a similar local loss surface (i.e. similar gradients)’ correspond to a model providing a similar explanation for those different examples.
>
>
> >I
>
> We added a small clarification (in bold) to the first sentence of the selective retrieval hypothesis paragraph where we use the word “rely”: *>the model learns to rely on them [define-dot definitions] more **(e.g. retrieve them more frequently)***.
>
>
> >J
>
> Thanks a lot for these comments and suggestions! We are considering the following edits.
>
> (J1) We tweaked the sentence introducing meta-OCL and pointed to Figure 1b there: *>More surprisingly, we observe such a difference in internalization even for statements that are equally compatible with other training examples, i.e. statements about variables for which no questions appeared in the training set (Figure 1b).* We will edit this further as we agree that clarity here is especially helpful for understanding the rest of the paper.
>
> (J2) For meta-OCL, we believe Figure 2 results are referenced quite clearly in Section 2.4, and the subsequent sentences explain why this is meta-learning. We will add a similar sentence when referencing Figure 2 in Section 2.3, and briefly recap the description of OCL from the intro.
>
> (J3) We will slightly improve the notation by replacing $\hat{\mathtt{QA}}_4$ with $\mathtt{QA}_4^{\text{not replaced}}$ or similar, and $\mathtt{QA}_7$ with $\mathtt{QA}_7^{\text{unseen vars}}$ or similar. We will also point out the connection between {$\mathtt{\dot{D}}_1\mathtt{QA}_1$, $\mathtt{\bar{D}}_2\mathtt{QA}_2$, $\mathtt{\dot{D}}_5$, $\mathtt{\bar{D}}_6$} with the columns of Figure 1.

---

### Official Review · Reviewer_Rz8f · 2023-10-27

**Soundness:** 3 good
**Presentation:** 1 poor
**Contribution:** 2 fair
**Rating:** 5
**Confidence:** 2

**Summary:**

The paper describes a new phenomenon called meta-out-of-context learning (meta-OCL). OCL refers to the observation that a model performs better on questions that include variable names when these variables are defined consistently. Meta-OCL refers to the observation that this also holds for variables for which no questions appeared in the training set. The authors demonstrate meta-OCL in various settings and offer two hypotheses for its emergence.

**Strengths:**

The effect is shown with different data sets, different models, and different settings. It therefore seems to be quite robust.

The effect is novel and has not been studied previously (to the best of my knowledge).

Improving our understanding of how neural networks generalize is interesting and important. The addition of true and false definitions adds a nice twist to it.

**Weaknesses:**

I found the paper difficult to read and follow. While the writing in general is fine, I found the explanations quite convoluted. It was by far the paper that took me the longest to digest/review despite being not super technical. Because of that, I would not be surprised if I misunderstood several things that I am pointing out below. I apologize for the potentially subpar review.

First of all, I am not convinced whether meta-OCL framing is needed here – what does it add? Is it not enough to say that neural networks generalize differently for true and false statements and that this holds for both unseen questions and unseen definitions? What is meta- about this? In Figure 12, the authors show that they observe similar effects in a single-stage setting, indicating that appealing to meta-learning might not be needed.

The authors perform many versions of their experiments. In general, this is a good thing. However, I found the presentation a bit strenuous. In many places, an explanation is given about the setting, but then the reader is referred to the SI for the results. It would be better if a subset of these experiments were moved to the SI entirely, leaving more space to put the plots/results for the other experiments in the main paper.

The related work section felt like quite a stretch. I found the discussed work not very relevant to the present paper.

The authors discuss that “reproducing this phenomenon with data real LLMs are trained on is an important avenue for future work” which I agree with. Yet, the current discussion on this is quite speculative. Maybe more could be done in this direction.

**Questions:**

Figure 2 is labeled “Performance on in-distribution questions”. This is quite confusing, as the figure states that it measures performance on the validation subsets (which is in line with my understanding of the paper). Which is correct?

How does Figure 3 show in-context learning? It seems like a comparison to the data from Figure 2 would be needed for that.

Why is the performance in Figure 4a lower than in the main experiment? Shouldn't it approach Figure 2 for alpha → 1?

From my intuitive understanding, I would have thought that having a pretrained model is necessary for this effect to appear. Yet, the authors show in Section 3.1 that pretraining is not necessary. Would it be possible to further elaborate on why the authors believe pretraining is not necessary? Performance in these experiments also seems to be quite low (54%) for a two-alternative forced-choice task. Why is that the case?

---

> ### Author Response · Authors · 2023-11-21
> **Addressing weaknesses**
>
> Thank you for a thoughtful review and suggestions.
>
> ## Addressing weaknesses
>
> >Is it not enough to say that neural networks generalize differently for true and false statements and that this holds for both unseen questions and unseen definitions? What is meta- about this?
>
> The fact that neural networks generalize differently in our experiments  means that the model updates differently when trained on consistent-seeming and inconsistent-seeming datapoints in the second finetuning stage. This difference in updates is why we call this phenomenon "meta-learning". Our results can be explained without mentioning meta-learning (e.g. the "selective retrieval" hypothesis in Section 4); still, we believe that meta-learning is an interesting and helpful frame for understanding what's happening.
>
> >In Figure 12, the authors show that they observe similar effects in a single-stage setting, indicating that appealing to meta-learning might not be needed.
>
> Indeed it is harder to view single-stage results as meta-learning, since it's unclear if the effect there comes from just training on all the data jointly, or a meta-learning-like effect. We employ the two-stage setting to disentangle these two explanations.
>
> >In general, [many versions of the experiments] is a good thing. However, I found the presentation a bit strenuous. In many places, an explanation is given about the setting, but then the reader is referred to the SI/Appendix for the results.
>
> Would it be fair to characterize your concern as "the references to the Appendix are too frequent"? We are wondering whether having less granular references to parts of the appendix would be helpful (e.g. instead of two references in “varying model size and experiments with other models”, have one reference to a larger appendix section relevant for both).
>
> >The related work section felt like quite a stretch. I found the discussed work not very relevant to the present paper.
>
> We’d appreciate any suggestions for the works that would be helpful to cover, or comments on which discussed works you believe to be the least relevant. We could see how the gradient alignment part of related work could seem extraneous given that we do not study this hypothesis in depth – was this the part that led to your concern?

---

> ### Author Response · Authors · 2023-11-21
> **Answering questions**
>
> ## Answering questions
> >Validation data and “performance on in-distribution questions”
>
> Both descriptions are correct – we always evaluate using validation subsets. We use “in-distribution” as a description of the question types: in-distribution validation data has the same questions types as the training data, whereas the entity attribution questions are never present in the training data. We’ll make this clearer in the caption of Figure 2.
>
> >How does Figure 3 show in-context learning? It seems like a comparison to the data from Figure 2 would be needed for that.
>
> In-context learning is the result of the setup for Figure 3: in this experiment, definitions appear in the same prompt as the question which the model needs to answer, and so the model learns to use these definitions in the prompt to answer questions. Comparison with Figure 2a tells us that out-of-context learning is a weaker effect than in-context learning, since consistent definitions help less when the model was previously trained on them VS the definition being in the prompt. We’ll add a sentence about this to the paragraph on “comparison with in-context learning”.
>
>
> >Why is the performance in Figure 4a lower than in the main experiment?
>
> See our response to question 2.1 from reviewer zUXo about the same point. We will change the question type to be consistent with Figure 2b (right now it is not) and re-run the experiment to check whether something else is affecting performance.
>
>
> >Would it be possible to further elaborate on why the authors believe pretraining is not necessary?
>
> Neither hypothesis for explaining meta-OCL discussed in Section 4 relies on pretraining. Below we will attempt to formulate a version of the “selective retrieval” explanation for the set inclusion experiment.
>
> During the first finetuning stage:
> 1. The first few epochs result in middle-layer activations of each variable token containing information on the definition that was present in the training data – specifically, information about both the define-tag token and the number present in the definition. This is easiest to imagine for the (variable, tag, entity) definition order, where the model is trained to predict the tag and the number from the variable.
> 2. In the remaining few epochs, the model learns to use this information when answering questions. When predicting the answer, the model outputs the variable’s definition number IFF the variable’s define-tag is D-dot. (Note how this kind of selective retrieval/output mechanism can achieve low training loss AND is representable by a neural net.)
> 3. Realistically, the above two steps are intertwined and affect one another, and are also intermixed with the model just memorizing the training data without connecting definitions to questions, so the conditional output rule is not learned perfectly.
>
> During the second finetuning stage, middle-layer activations of new variable tokens end up being structured similarly: they also contain info on the define tag and the number that followed the variable in the definitions. The selective retrieval mechanism from the first finetuning stage is unchanged, and results in the meta-OCL effect.
>
>
> >Why is the effect size for the set inclusion experiment so small?
>
> In this experiment, presence & strength of OCL & meta-OCL depends a) number of entity-variable pairs in the training data, and b) number of QA examples per entity-variable pair. Increasing a) helps the model learn the task but also makes the task more difficult since the model needs to learn more variable binding relations. Increasing b) helps the model learn the task as well, but makes definitions less useful as the model can infer more from the QA pairs.
>
> Consistently with this, we found that when we increase the number of QA examples per entity-variable pair, the first stage performance becomes very good (it’s easy to push it to almost 100% accuracy), but any effect in the second stage disappears.
>
> We believe it should be possible to tune these and other parameters (such as how many integers are present in each question) and attain a stronger effect; still, our experiments are sufficient to show that pretraining is not needed for meta-OCL.

---

> > ### Comment · Reviewer_Rz8f · 2023-11-22
> >
> > Thanks to the authors for their reponse. I will not have the time to go through it today. I'll read it later this week and update here again if possible.

---

### Official Review · Reviewer_UZ6L · 2023-10-31

**Soundness:** 3 good
**Presentation:** 2 fair
**Contribution:** 2 fair
**Rating:** 5
**Confidence:** 4

**Summary:**

This paper analyzes the language model (LM) behavior on the learned prior when the in-context samples (from the learned dataset) are (i) consistently true or (ii) consistently false. When the in-context samples are consistently true, the LM more rely on the learned prior, while in the opposite case, the LM more rely on the in-context information.

**Strengths:**

1. The phenomenon itself is interesting.

2. The overall experiment is rigorous and well-defined.

3. Considered various model size for the analysis which is important to explain the phenomenon

**Weaknesses:**

(1) The overall writing can be much improved.
- The paper introduces new terminology without defining them in the first place, e.g., meta-out-of-context learning (in the abstract), internalize, definitions. Especially, the abstract is not understandable before reading the main text.
- Also, the mathematical definitions of the terms are missing, or rigorous quantification will be helpful, e.g., quantifying internalize.
- I think the following sentence is too general: language models trained with gradient-descent-based methods. Most of the existing language models use gradient descent to train, including RNNs and LSTMs, while this paper focuses on recent language models.
- The Subset definition in Table 1 is quite complicated. When reading the results, I have to read Table 1 multiple times to understand the results.

(2) Lack of explanation of why such a phenomenon happens. I think the analysis should be quite different from in-context learning as in-context learning is mainly about generalizing on a novel task [1], but this is mainly about the seen task (the known knowledge).

(3) I can not see the actual used case of out-of-context learning. For instance, in-context learning is used as an adaptation of language models w.o parameter updating.

(4) It will be interesting to see the comparison on aligned LLMs, i.e., the instruction finetuned LLMs (e.g., Llama-chat) [2], as these models behave differently for in-context samples [3].

Reference\
[1] Transformers learn in-context by gradient descent, ICML 2023\
[2] Llama 2: Open Foundation and Fine-Tuned Chat Models, arXiv 2023\
[3] Larger language models do in-context learning differently, arXiv 2023

**Questions:**

Written in the weakness part.

---

> ### Author Response · Authors · 2023-11-21
>
> Thanks for your review! We are glad you found the phenomenon we study interesting and our experiments sound. We address the weaknesses below.
>
> 1\. Improving clarity
> - Re abstract: we are considering replacing the second sentence of the abstract with
>
>     >*Using carefully designed synthetic experiments with LLMs, we establish the existence of a phenomenon we call meta-out-of-context learning (meta-OCL): LLMs learning how to update on new datapoints in a way helpful for generalization but not explicitly encouraged by the training loss.*
>
>     Would such a change address your concern about the abstract’s readability?
>
> - We believe that while our current operationalization of internalization (“model treats a statement as true when generating new text”) can be formalized somewhat, trying to write this down in math would not make our paper clearer. For example, we could say that statement A is epsilon-internalized when “generated statements implying NOT A are epsilon times as frequent as generated statements that do imply A”. However, this is still very abstract (and requires an oracle that can tell whether one statement implies another), and so we believe it's best to leave the current informal operationalization as is.
>
>
>     We think there are other promising directions to study meta-OCL more formally, such as formalizing a datapoint's usefulness for modeling other data (see our response to reviewer zUXo's Q1.1).
>
>
> - Our experiments with transformers cover both decoder-only and encoder-decoder architectures, as well as both finetuning these models and training them from scratch. We also present an experiment showcasing meta-OCL with a ConvNet. We would be quite surprised if it was not possible to reproduce meta-OCL with LSTMs or RNNs – do you expect otherwise? Given this, we believe it is fair to claim the quoted level of generality, as opposed to e.g. making that claim only about transformer-based language models.
>
> - We agree that our setup is quite involved, and would like to improve our description of  it. In response to your comment we will modify the caption of Table 1 to point out the connection between {$\mathtt{\dot{D}}_1\mathtt{QA}_1$, $\mathtt{\bar{D}}_2\mathtt{QA}_2$, $\mathtt{\dot{D}}_5$, $\mathtt{\bar{D}}_6$} data subsets and the columns of Figure 1. See also our response to reviewer zUXo about potential improvements to our notation – we’d appreciate any thoughts you might have on this.
>
>
>
> 2\. We agree that our two hypotheses for explaining meta-OCL (selective retrieval and gradient alignment, both in Section 4) are far from being conclusive explanations; we note this in the Limitations part of Section 6. Since the phenomenon we study is quite different from in-context learning, our analysis is very different as well – are you saying it is similar? We discuss the relationship between out-of-context and in-context learning in the corresponding paragraph of the related work section.
>
>
> 3\. The primary usecase of our work is to better understand what kinds of learning LLM are capable of. We do point out several implications of meta-OCL in Section 6, specifically that this capability could be part of a mechanism for how LLMs might acquire situational awareness, as well as learn skills such as using a sophisticated decision theory from just being trained on descriptions of such skills. As for more practical usecases, we believe a better understanding of meta-OCL is necessary before it would be possible to say that e.g. one can instill some desirable property into a LLM by leveraging this capability.
>
>
> 4\. We are not familiar with setups that involve finetuning a LLM in a self-supervised fashion after it was already finetuned with RLHF, which is why we did not include this kind of experiment. Following your comment we did run this experiment with `Llama-2-7b-chat-hf` out of interest, and the results are very similar to what we report for regular `Llama-2-7b` in Figure 16.

---

### Official Review · Reviewer_zUXo · 2023-11-01

**Soundness:** 2 fair
**Presentation:** 2 fair
**Contribution:** 3 good
**Rating:** 6
**Confidence:** 2

**Summary:**

This paper showcases a learning effect where specific operators are introduced to perform variable binding in either a way that is congruent/predictive for other datapoints (e.g. Q/A pairs), or incongruent. Using a congruent operator (\dotted{Define}) makes the model more accurate at recalling information about these variables, whereas the incongruent operator (\bar{Define}) hurts performance. This effect is seen also when using held-out datapoints (e.g. new Q/A pairs)
The authors interpret this as evidence for a new concept of “out-of-context” and “meta-out-of-context” learning, placing it in contrast with recent “in-context learning” effects from LLM.
They show extensive experiments using LLMs on QA datasets, early experiments using visual modalities and discuss potential implications.

Overall, I found this work interesting but I have reservations as my stance is entirely summarized by their section on “The model learns the semantics of the define tags correctly”, and I feel like this is “just the model learning about predictive correlations in the data” and I do not find any of the results especially surprising.

However this paper feels controversial and novel enough that it may deserve a larger discussion within the community (to either prove or disprove its assumptions / observations), so I will lean towards acceptance

**Strengths:**

1. The paper is very thorough in presenting its arguments, observations and interpretations. It does a good enough job (after one has read far enough) to introduce the complex interplay between data subsets (e.g. Table 1), how things were designed and trained.
2. Results, figures and ablations for all the LLM sections are strong.
3. It was very useful to have the alternative interpretation at the end of Section 4. This was exactly what I wanted to raise, and this will make this discussion more fruitful (but I probably would have preferred to have this come earlier in the paper)
4. The paper discusses mechanisms and implications, which goes beyond “just” reporting a new finding. The paper does feel like it has been iterated on several times.

**Weaknesses:**

1. I found the paper rather confusing at first, and quite hard to understand. The desire to introduce new terms and to use concepts such as “internalization” was counterproductive. Figure 1 ended up being useful to understand what was done, but it took me quite a while longer to fully grasp it as I would have liked. The choice to stick to the out-of-context framing felt counterintuitive to me, instead of being anchored in a more classical “learning the data” theory. Talking about “variable binding” earlier would have helped.
2. The experiments in Section 3 felt significantly weaker than the rest. All effects in the Appendix were weak (i.e. gap between congruent vs incongruent operators), and the MNIST-based visual task was quite strained.
3. Despite how the abstract made it look like, I did not find the discussion with the safety implications particularly deep or illuminating. The functional decision theory argument was not clear enough to me.
4. As a small issue: the exact choice of notation for the operators wasn’t very helpful (\dot{Define} and \bar{Define}), purely from how similar they look. Similarly, the data subsets were quite hard to follow and you could have used more varied letters to help the reader.

**Questions:**

1. As explained above, my interpretation of the results shown are still entirely in the camp of “the model learnt two variable binding operators”. I am not convinced by the arguments presented at the end of section 4 about how this does not fully predict the results.
   1. Consider replacing all arguments about “truthfulness” with “predictability”, would you still keep the same arguments throughout the paper? I personally wouldn’t.
   2. “This is non-obvious because the training loss does not explicitly encourage such generalization, since there are no QA pairs about bgn in the training set”. This is not what is happening. The model learnt that the congruent-Define is indeed a variable binding from bgn to Darwin, and hence will then replace all potentially new QA/data pairs about bgn with Darwin and behave as such. This will happen even in novel situations, through the model activations? If the argument is about “how many updates does it take for the congruent-Define to be learnt”, I’m not sure this can be appropriately assessed (the datasets are small, you perform many epochs, Transformers are big, etc).
   3. However, the incongruent-Define was trained to be non-predictive, and in effect is probably learning to map bgn to a “zero” embedding (akin to what people are doing for classifier-free guidance), which has low correlation with all other entities in the dataset. It behaves entirely differently to congruent-Define, and hence trying to compare them is like trying to compare how predictable different datapoints are: it depends on the dataset, and if they aren’t predictable accuracies will be lower. You cannot really use its failure as a contrast to the “success” of congruent-Define in being effectively learnt.
2. In practice, I think Figure 4a is the crux of the issue and what would shine light on exactly what is happening
   1. First of all, why are exact match number so much lower than in Figure 2? I would have expected alpha -> 1 to match D1 in Figure 9d, so ~0.2? The fact that you changed the question type is also unfortunate, using “What is the name” and referring directly to Figure 2b would be more helpful. You also do not explicitly say that the comparison is Figure 2b (it could have been 2a?).
   2. Changing alpha is indeed modifying the “predictibility” of a definition, and with it how “useful” the congruent-operator is.
   3. Alpha controls the performance exactly as I expect it to, but I would have wanted to see values for alpha < 0.5 however to see how low it gets.
   4. The issue is that you made the fraction of inconsistent entities for the incongruent-operator *equal* to this new number of consistent entities. For alpha=0.75, this implies that you also have a large number of inconsistent entities for the incongruent-operator. And so in Figure 4a, the D9 number is fully explained by “you just have more data that shows that the incongruent-operator is a random map”, it goes down in the same way as it does in Figure 2.
   5. Could you please explain what one should expect to see differently to counter the interpretation above?
3. I understand that notation is difficult, but I would like to flag again that I found most of it confusing, even while writing this up (i.e. operators, datasets, data subsets, stages, most of them). There are many modifiers applied to very few symbols, instead of a small clearer set of well-defined and informative letters/words.
4. The fact that question types are shared across datapoints feels potentially problematic? (Appendix A.3). Don’t they provide correlations for the model to hook onto, regardless of variables?

---

> ### Author Response · Authors · 2023-11-21
> **Addressing weaknesses**
>
> Thank you for a thoughtful and constructive review.
>
> ## Addressing weaknesses
>
> 1. We will mention the variable binding explanation ("the model learns the semantics of the define tags correctly" from Section 4) in the last paragraph of the introduction, which will hopefully make our paper clearer for future readers.
>
> 2. Regarding experiments from section 3 (set inclusion and MNIST): any systematic difference, even if small, is a positive result here. We do agree that our MNIST setup is quite far from any real-life computer vision setting; it is however very similar to the set inclusion setup, and the presence of meta-OCL there indicates that this phenomenon is not limited to transformers. (You might also be interested in our response to question 4 from reviewer Rz8f about the factors affecting set inclusion experiment's performance.)
>
> 3. Your comment closely mirrors points E and F from reviewer Vaqj. Following these comments, we plan to expand the paragraph on functional decision theory and move it to the appendix, and add a sentence clarifying the connection of our work to situational awareness.
>
> 4. Regarding notation: we agree our notation is quite involved, and would like to make it clearer and easier to understand. One small improvement is to replace $\hat{\mathtt{QA}}_4$ with $\mathtt{QA}_4^{\text{not replaced}}$ or similar, and $\mathtt{QA}_7$ with $\mathtt{QA}_7^{\text{unseen vars}}$ or similar. We’d also love to improve on $\mathtt{\dot{D}}$/$\mathtt{\bar{D}}$, but we don’t have good ideas for how to do it given that we also want to keep both the subscripts (for data subset number) and the superscripts (for whether a definition is consistent). We’d appreciate any suggestions you might have here.

---

> ### Author Response · Authors · 2023-11-21
> **Answering questions**
>
> ## Answering questions
>
> 1.1. Replacing the arguments about “truthfulness” with “predictability” preserves the meta-learning framing: in the second finetuning stage, the model internalizes datapoints that look like they might be more predictive of other data (but the model never saw such data of which these datapoints are predictive). We believe formalizing this mathematically is a good avenue for future studies of meta-OCL; a metric similar to predictability that we think is highly relevant is something like “how much does retrieving knowledge from a given datapoint helps reduce the training loss on other training data”, which is also similar to Data Shapley [1].
>
> 1.2. We do not make any points about “how many updates does it take for the congruent-Define to be learnt”. Rather, we claim that the fact that the model learns to bind the variable bgn to the entity Darwin from being trained on the string “congruent-Define bgn Darwin” can be seen as meta-learning, and would be surprising to a substantial fraction of the ML community. We believe the ability of LLMs to internalize semantic content of the text (as opposed to shallow token co-occurrence statistics) is currently under-appreciated, and our study is an attempt to highlight and better understand this ability.
>
> 1.3. We might be missing your argument here: indeed in $\mathcal{X}_1$ (first stage finetuning data) define-dot definitions are predictive of other training datapoints, while define-dash definitions are much less predictive. This is not the case in $\mathcal{X}_2$ – here both types of definitions are equally predictive. The purpose of comparing the two is to demonstrate that the model learns to update on datapoints resembling those that were more/less predictive in the past in a way advantageous for generalization.
>
> 2.1. Thanks for catching this – we’re not sure why the numbers are so different, indeed it is a bit surprising that the last 5% of $\alpha$ should make as much of a difference. We are re-running this experiment to see if we can catch the issue. We will also change the question type and point out the connection with Figure 2b explicitly, thanks!
>
> 2.2. Agreed!
>
> 2.3. A setting with a given $\alpha$ is equivalent to that with 1-$\alpha$, so we don’t expect $\alpha$<0.5 experiments to be different from the ones presented in Figure 4a. The only difference is $\mathtt{\dot{D}}$ and $\mathtt{\bar{D}}$ changing places in which one is the more consistent/predictive one; in our case the define tags are random strings that are re-sampled across multiple seeds, so it does not matter which one we call $\mathtt{\dot{D}}$ and which $\mathtt{\bar{D}}$.
>
> 2.4. and 2.5. We did not understand the sentence below and would appreciate clarifications:
> >And so in Figure 4a, the D9 number is fully explained by “you just have more data that shows that the incongruent-operator is a random map”, it goes down in the same way as it does in Figure 2.
>
> We believe the $\mathtt{\bar{D}}^\text{consis}_9\mathtt{QA}_9$ result is interesting because for high $\alpha$, the define tag pushes the model to internalize definitions *less* (since, as you say, there is enough data to learn the define-incongruent meaning), while the actual definition consistency pushes the model to internalize definitions *more* (internalizing specifically these definitions helps predict the answers to the training questions). This setting is different from that in the first stage of Figure 2, where these two factors always push in the same direction. It is also different from the second stage of Figure 2, where only the define tag affects internalization (since there are no training QA pairs for the model to tell whether the definitions are in fact consistent).
>
> 3\. See our comment regarding notation on Weakness 4.
>
> 4\. Question type sharing does indeed provide base rates for the answers. E.g. for a test question “what did xyz do” the model’s answer would be informed not just by the definition and the QA pairs about xyz, but also by the answers to this type of question for other variables in the training data. This is not an issue for us since our results are all based on the relative differences between the various data subsets.
>
> [1] Ghorbani, Amirata, and James Zou. "Data shapley: Equitable valuation of data for machine learning." International conference on machine learning. PMLR, 2019.

---

### Meta-Review · Area_Chair_ALTL · 2023-12-07

**Metareview:**

This paper provides extensive experiments studying the effect of consistent vs inconsistent variable binding training data in model’s accuracy on Q/A setups about said entities. Reviewers have unanimously found experiments to be thorough, with various ablations, across several models and datasets. The paper indeed provides lots of interesting experimental setups, and has great potential.

However, the presentation lacks clarity. Reviewers have found notation and terminology confusing, and not properly defined before usage. Formal definition and rigorous quantification seem to be missing. For example, several reviewers have argued that “internalization” is ill-defined, and the main phenomena the paper discusses (OCL and meta-OCL) need to be formalized more clearly. Otherwise, the experimental results risk being summarized as “another expected pattern matching behavior”. There are concerns about whether meta-OCL is the right framing for explaining the experimental results.

The reviewers have provided lots of great suggestions to improve presentation of the work and there has been significant and constructive engagement during the discussion period. However, the draft has not been updated to address these most pressing concerns about formalization and clarity. Therefore, I recommend another round of revision to incorporate the excellent suggestions provided by the insightful reviews and to clearly convey the insights from these extensive and interesting experiments. This manuscript would significantly benefit from major structural changes to conceptual formalization and presentation.

**Justification For Why Not Higher Score:**

The presentation of the work is not clear, which is detrimental to communicating the findings. Additionally, there are concerns about clearly formalizing the core concepts the paper discusses, and addressing them is required before publication.

**Justification For Why Not Lower Score:**

N/A

---

### Decision · Program_Chairs · 2024-01-16

Reject